



**Winter observations of ClNO₂ in northern China: Spatiotemporal variability and**
**insights into daytime peaks**
Men Xia[1], Xiang Peng[1], Weihao Wang[1,8], Chuan Yu[1,2], Zhe Wang[6], Yee Jun Tham[7],
Jianmin Chen[4], Hui Chen[4], Yujing Mu[5], Chenglong Zhang[5], Pengfei Liu[5], Likun Xue[2],
Xinfeng Wang[2], Jian Gao[3], Hong Li[3], and Tao Wang[1]
[1]Department of Civil and Environmental Engineering, The Hong Kong Polytechnic
University, Hong Kong SAR, China
[2]Environment Research Institute, Shandong University, Ji'nan, Shandong, China
[3]Chinese Research Academy of Environmental Sciences, Beijing, China
[4]Department of Environmental Science and Engineering, Fudan University, Institute of
Atmospheric Sciences, Shanghai, China
[5]Research Center for Eco-Environmental Sciences, Chinese Academy of Sciences,
Beijing, China
[6]Division of Environment and Sustainability, Hong Kong University of Science and
Technology, Hong Kong SAR, China
[7]Institute for Atmospheric and Earth System Research/Physics, University of Helsinki,
Helsinki, Finland
[8]Hangzhou PuYu Technology Development Co., Ltd, Hangzhou, Zhejiang, China
Correspondence: Tao Wang (cetwang@polyu.edu.hk)
**Abstract**
Nitryl chloride (ClNO₂) is an important chlorine reservoir in the atmosphere that affects
the oxidation of volatile organic compounds (VOCs) and the production of RO$_x$ radicals
and ozone (O₃). This study presents measurements of ClNO₂ and related compounds at
urban, rural, and mountain sites in the winter of 2017–2018 over the North China Plain
(NCP). The nocturnal concentrations of ClNO₂ were lower at the urban and rural sites
but higher at the mountain site. The winter concentrations of ClNO₂ were generally
lower than the summer concentrations that were previously observed at these sites,
which was due to the lower nitrate radical (NO₃) production rate ($P$(NO₃)) and the
smaller N₂O₅ uptake coefficients ($\gamma$(N₂O₅)) in winter, despite the higher dinitrogen
pentoxide (N₂O₅) to NO₃ ratios in winter. Significant daytime peaks of ClNO₂ were
observed at all the sites during the winter campaigns, with ClNO₂ mixing ratios of up
to 1.3 ppbv. Vertical transport of ClNO₂ from the residual layers and prolonged
photochemical lifetime of ClNO₂ in winter may explain the elevated daytime
concentrations. The daytime-averaged chlorine radical (Cl) production rates ($P$(Cl))
from the daytime ClNO₂ were 0.17, 0.11, and 0.12 ppbv h$^{-1}$ at the rural, urban, and
mountain sites, respectively, which were approximately 3–4 times higher than the
campaign-averaged conditions. Box model calculations showed that the Cl atoms
liberated during the daytime peaks of ClNO₂ increased the RO$_x$ levels by up to 27–37 %
and increased the daily O₃ productions by up to 13–18 %.

**Key points:**
1. Winter measurements of ClNO₂ concentrations were made at rural, urban, and



mountain sites in northern China.
2. The elevated daytime mixing ratios of ClNO$_2$ were up to 1.3 ppbv.
3. The daytime peaks of ClNO$_2$ increased the concentration of ROx radicals by up to
27–37 % and the net O$_3$ production by 13–18 %.

**1. Introduction**


Cl is a potent atmospheric oxidant that reacts analogously to hydroxyl radicals (OH)
with hydrocarbons (Wang et al., 2019b). Cl is highly reactive toward alkanes, with the
rate constants of its reactions with alkanes being approximately 10–200 times greater
than some of the OH + VOCs reactions (Atkinson and Arey, 2003; Burkholder et al.,
2015). Consequently, Cl enhances the production of RO$_x$ (= OH + HO$_2$ + RO$_2$) via
Reactions R1-R4, which promotes O$_3$ formation by converting nitric oxide (NO) to
nitrogen dioxide (NO$_2$) (Reactions R3 and R5). Cl also consumes O$_3$ via Reaction R8.
The net effect of the reactivity of Cl is typically the depletion of O$_3$ in the stratosphere
(Molina and Rowland, 1974) and an increase in O$_3$ production in the polluted
troposphere (Riedel et al., 2014; Xue et al., 2015).
(R1) RH$(g)$ + Cl$(g)$ → R$(g)$ + HCl$(g)$
(R2) R$(g)$ + O$_2(g)$ + M → RO$_2(g)$ + M
(R3) RO$_2(g)$ + NO$(g)$ → RO$(g)$ + NO$_2(g)$
(R4) RO$(g)$ + O$_2(g)$ → HO$_2(g)$ + products
(R5) HO$_2(g)$ + NO$(g)$ → OH$(g)$ + NO$_2(g)$
(R6) NO$_2(g)$ + $hv$ → NO$(g)$ + O($^3$P)
(R7) O$_2(g)$ + O($^3$P) + M → O$_3(g)$ + M
(R8) Cl$(g)$ + O$_3(g)$ → ClO$(g)$ + O$_2(g)$
where M denotes the third body in ambient air.

The production of Cl is determined by the formation and decomposition of Cl
precursors such as ClNO$_2$ (Chang et al., 2011). ClNO$_2$ is produced mostly in dark
conditions by the heterogeneous uptake of N$_2$O$_5$ on chloride (Cl$^-$)-laden aerosols
(Reactions R9–R13) and undergoes photolysis during the day (Reaction R14)
(Finlayson-Pitts et al., 1989). ClNO$_2$ formation is constrained by the NO$_3$ production
rate ($P$(NO$_3$), Reaction R9). NO$_3$ is in thermal equilibrium with N$_2$O$_5$ (Reaction R10),
and the equilibrium constant (K$_{eq}$) depends on temperature and NO$_2$ concentrations.
N$_2$O$_5$ formation is suppressed by NO and VOCs as they consume NO$_3$ (Reactions R11–
12). The N$_2$O$_5$ uptake probability ($\gamma$(N$_2$O$_5$)) and ClNO$_2$ production yield ($\varphi$(ClNO$_2$))
are kinetic parameters with values between 0 and 1, which can be derived from the
observation data of N$_2$O$_5$, ClNO$_2$, and related species (Brown et al., 2006; Phillips et
al., 2016). Previous laboratory studies have demonstrated that $\gamma$(N$_2$O$_5$) is enhanced by
higher relative humidity (RH) and particulate Cl$^-$ concentrations but suppressed by
higher temperature and concentrations of aerosol nitrate (NO$_3^-$) and organic species
(Behnke et al., 1997; Hallquist et al., 2003; Bertram et al., 2009; Griffiths and Anthony
Cox, 2009).
(R9)                    NO$_2(g)$ + O$_3(g)$ → NO$_3(g)$ + O$_2(g)$
(R10)                   NO$_3(g)$ + NO$_2(g)$ ↔ N$_2$O$_5(g)$


| 88 | (R11) | $NO_3(g) + NO(g) \leftrightarrow 2NO_2(g)$ |
| 89 | (R12) | $NO_3(g) + VOCs(g) \rightarrow products$ |
| 90 | (R13) | $N_2O_5(g) + Cl^-(aq) \rightarrow ClNO_2(g) + NO_3^-(aq)$ |
| 91 | (R14) | $ClNO_2(g) + hv \rightarrow Cl(g) + NO_2(g)$ |

Field observations of $ClNO_2$ were first reported in the marine boundary layer off the coast of the Houston-Galveston area in the USA (Osthoff et al., 2008). Subsequent studies demonstrated the worldwide ubiquity of $ClNO_2$ and confirmed its significant role in photochemistry (Thornton et al., 2010; Mielke et al., 2011; Phillips et al., 2012; Edwards et al., 2013; Bannan et al., 2015; Wild et al., 2016; Wang et al., 2016; Bannan et al., 2019; Eger et al., 2019). The role of $ClNO_2$ in the radical budget could be more important than that of OH in winter, because OH production is reduced in winter owing to lower concentrations of $O_3$ and $H_2O$ vapor in this season (Haskins et al., 2019). A limited number of winter observations of $ClNO_2$ have been conducted on various platforms, including on aircrafts over northern Europe (Priestley et al., 2018) and the eastern US (Haskins et al., 2019), on a tall tower in Boulder, USA (Riedel et al., 2013), on a mountain top in Hong Kong (Wang et al., 2016), and at ground sites in Alberta, Canada (Mielke et al., 2016) and Heshan, China (Yun et al., 2018). These studies found high $ClNO_2$ mixing ratios of up to 7.7 ppbv (Yun et al., 2018) in winter and a contribution of $ClNO_2$ to Cl liberation of up to 83 % (Priestley et al., 2018).

The chemical production of $ClNO_2$ in winter has some unique features compared with that in warmer seasons. Long winter nights provide more time for $ClNO_2$ production and accumulation. Lower temperatures in winter shift the $N_2O_5$-$NO_3$ equilibrium to the $N_2O_5$ side (Brown et al., 2003) and increase the $\gamma(N_2O_5)$ on aerosols (Bertram and Thornton, 2009). However, $P(NO_3)$ might be lower in winter due to reduced $O_3$ concentrations. The availability of aerosol $Cl^-$ also varies in winter and summer. The winter monsoon brings air masses from the interior of the continent, thereby suppressing the transport of sea salt to inland areas. However, more $Cl^-$ is emitted due to coal burning in winter (McCulloch et al., 1999; Fu et al., 2018). Thus, considering the complexity of $N_2O_5$ chemistry and $Cl^-$ sources, it is not clear whether $ClNO_2$ formation is more prevalent in winter.

The North China Plain (NCP) – home to Beijing and several other megacities – is one of the most industrialized and populous regions of China, and frequently suffers from severe haze pollution in winter (An et al., 2019; Fu et al., 2020). $ClNO_2$ concentrations have been measured over the NCP (Breton et al., 2018; Zhou et al., 2018), but only one study was conducted in winter (Breton et al., 2018). This study presents recent field observations of $ClNO_2$ concentrations from three campaigns conducted in winter and early spring at three sites in the NCP. The results were compared with those obtained in the previous summer campaigns at the same locations. We examined the factors controlling $ClNO_2$ formation, i.e., $P(NO_3)$, the nocturnal reactivity of $NO_3$ and $N_2O_5$, $\gamma(N_2O_5)$, and $\varphi(ClNO_2)$. We then focused on the unexpected daytime peaks of $ClNO_2$ concentrations that were observed at the sites and





evaluated their impact on the daytime atmospheric oxidative capacity using a chemical
box model.
**2. Methods**
2.1 Observation sites
Field campaigns were performed in Wangdu, Beijing, and Mt. Tai in sequence during
the winter-early spring of 2017–2018 (Table 1). The locations of the three sites are
shown in Fig. S1. The sites were selected for investigation of ClNO$_2$ in urban, rural,
and mountain areas of the NCP. The winter indoor-heating period lasts from early
November to 15 March of the following year (Ran et al., 2016), and thus the
observations were made mostly during the heading period during which coal is
intensively used. Detailed descriptions of the measurement sites are available in
previous studies (Tham et al., 2016; Wang et al., 2017c; Xia et al., 2019), and a brief
introduction is given here.
Table 1. Locations and periods of the field campaigns relevant to this study.

| Location | Site category | Season | Observation period | Coordinate |
|---|---|---|---|---|
| Wangdu | Rural | Winter[1] | 9-31 December 2017 | 38.66° N, 115.25° E |
| | | Summer[2] | 21 June to 9 July 2014 | 38.67° N, 115.20° E |
| Beijing | Urban | Winter[1] | 6 January to 1 February 2018 | 40.04° N, 116.42° E |
| | | Early summer[3] | 24 April to 31 May 2017 | |
| Mt. Tai | Mountain | Winter to early spring[1] | 7 March to 8 April 2018 | 36.25°N, 117.10°E |
| | | Summer[4] | 24 July to 27 August 2014 | |

[1]Observations from this study.
[2-4]Observations from previous studies, i.e., Tham et al. (2016), Xia et al. (2019), and
Wang et al. (2017c), respectively.
Our observations at the Wangdu site were part of the Campaign of Oxidation
Potential Research for air Pollution in winter (COPPER). The Wangdu site is located in
Dongbaituo Village, Hebei Province. Local villagers use coal stoves for cooking and
heating during winter. National road G4 and provincial road S335 are 1 km and 3 km
to the west of the sampling site, respectively. Many heavy-duty trucks passed through
G4 and S335 every night during the study period, emitting a large amount of NO$_x$ and
particulate matters. Therefore, the site experienced heavy pollution from coal burning
and road traffic (Peng et al., 2020).
The Beijing site is located at the Chinese Research Academy of Environmental
Science (CRAES), which is 15 km northeast of the city center. The sampling site is
surrounded by intra-city roads, commercial buildings, and residential buildings with





few industrial facilities. When the prevailing wind originates from the north (i.e.,
remote mountainous regions), the site is upwind from the Beijing downtown area and
thus is less polluted. However, when the wind originates from the south, the site
receives pollutants from Beijing's urban areas in the NCP (Xia et al., 2019).

Mt. Tai is located approximately 40 km south of Jinan City (population: 8.9 million)
and 15 km north of Tai'an City (population: 5.6 million) (Wen et al., 2018).
Measurements were taken at Mt. Tai meteorological station (1534 m a.s.l.). The site is
isolated from the anthropogenic emissions of tourist areas and is not affected by local
emissions. The observation period, i.e., March to April, was in early spring in the NCP.
However, considering the low temperature (4.6 ± 6.3 ℃) observed on top of Mt. Tai,
this study considered the observation period as winter to early spring.

2.2 Measurements of $N_2O_5$ and $ClNO_2$ concentrations
$N_2O_5$ and $ClNO_2$ were simultaneously measured by a chemical ionization mass
spectrometer with a quadrupole mass analyzer (Q-CIMS; THS Instruments). The
principles and calibrations of the Q-CIMS measurements are available in previous
studies (Tham et al., 2016; Wang et al., 2017c; Xia et al., 2019). The primary ions used
in the Q-CIMS were iodide ($I^-$) and its water clusters, which were generated using $CH_3I$
with an inline ionizer ($^{210}Po$). The iodide adducts, namely $IN_2O_5^-$ and $IClNO_2^-$, were
then detected by the mass spectrometer. An example of the mass spectrum is shown in
Fig. S2. The isotopic ratios of $I^{35}ClNO_2^-$ and $I^{37}ClNO_2^-$ in the ambient data were used
to confirm the identity of $ClNO_2$ (Fig. S3). Gas-phase mixtures of $NO_2$ and $O_3$ produced
$N_2O_5$ in a dynamic gas calibrator (Sabio Instruments) for $N_2O_5$ calibration. The
synthetic $N_2O_5$ was converted to $ClNO_2$ by passage through a humidified NaCl slurry
for $ClNO_2$ calibration. On-site calibrations were performed every 1–2 days, and
background detections of $N_2O_5$ and $ClNO_2$ were conducted every day by passing
ambient air through glass wool. The dependence of the $N_2O_5$ sensitivities (normalized
to the signal of $I(H_2O)^-$) on ambient RH was tested and used to calibrate the $N_2O_5$ data
(Fig. S4). The normalized sensitivity of $N_2O_5$ is the signal ratio of $I(N_2O_5)^-$ to $I(H_2O)^-$
in the presence of 1 pptv of $N_2O_5$. The normalized sensitivities and detection limits of
the $N_2O_5$ and $ClNO_2$ measurements were $(0.9–2.2) \times 10^{-5}$ Hz/Hz/pptv and 4–7 pptv ($3\sigma$
in 5 minutes), respectively during the three campaigns. The variation in the sensitivities
and detection limits of $N_2O_5$ and $ClNO_2$ were small within each campaign (Text S1,
Table S1, and Fig. S5). A virtual-impactor design (Peng et al., 2020) was adopted, and
the sampling tube was replaced daily to minimize inlet artifacts.

2.3 Other measurements
The trace gases, particle number size distribution (PNSD), ionic composition of
aerosols and other species were simultaneously measured (Table S2). Online VOCs
measurements were performed by gas chromatography-flame-ionization
detection/mass spectrometry (GC-FID/MS; Chromatotec Group) at the Beijing site
(Zhang et al., 2017) and Wangdu site (Zhang et al., 2020). At Mt. Tai, we used canisters
to collect air samples, which were analyzed using GC-FID/MS. The ionic compositions





of $PM_{2.5}$ (e.g., $NH_4^+$, $NO_3^-$, $SO_4^{2-}$, and $Cl^-$) were quantified by the Monitor for AeRosols
and GAses in ambient air (MARGA, Metrohm) at the Beijing and Mt. Tai sites (Wen
et al., 2018). An aerosol chemical speciation monitor (ACSM, Aerodyne Research Inc.)
was utilized at the Wangdu site to monitor the non-refractory components of these ions.
The concentrations of the $NO_3^-$, $SO_4^{2-}$, and $NH_4^+$ measured simultaneously by the
MARGA and ACSM were in good agreement, whereas the concentration of $Cl^-$
measured by the ACSM was slightly lower than that measured by the MARGA, which
was possibly due to the significant proportion of refractory chloride, e.g., NaCl, present
in the aerosols (Xia et al., 2020). We assumed that the particles sampled by a wide-
range particle spectrometer (WPS) were spherical in shape and calculated the aerosol
surface area density ($S_a$) and volume density ($V_a$). A parameterization was adopted to
consider the hygroscopic growth factor (GF) of aerosol sizes, as follows:
$GF = a \times \left(b + \frac{1}{1\text{-RH}}\right)^{1/3}$ (Lewis, 2008), where the parameters a and b were derived as
0.582 and 8.460, respectively in a previous field study over the NCP (Achtert et al.,
222   2009).


2.4 Calculation of $N_2O_5$ loss and $ClNO_2$ production
Some analytical metrics were estimated from the observation data. $P(NO_3)$ was
calculated using Eq. (1), where $k_1$ represents the rate constant of Reaction R9.
(Eq. 1) $P(NO_3) = k_1 \times [O_3] \times [NO_2]$
$k(NO_3)$ during the night was estimated using the measured mixing ratios of NO and
VOCs.
(Eq. 2) $k(NO_3) = \sum k_i[VOC_i] + k_{NO+NO_3}[NO]$
where $k_i$ is the rate constant for a specific VOC + $NO_3$ reaction and $k_{NO+NO_3}$ represents
the rate constant for Reaction R11. The ambient concentrations of $NO_3$ were estimated
by assuming that $NO_3$ and $N_2O_5$ were in dynamic equilibrium.
(Eq. 3)  $[NO_3] = \frac{[N_2O_5]}{[NO_2]K_{eq}}$
The loss rates of $NO_3$ due to NO and VOCs were then calculated by $k_{NO+NO_3}[NO][NO_3]$
and $\sum k_i[VOC_i][NO_3]$, respectively.
The loss rate coefficient of $N_2O_5$ on the aerosol surface ($k(N_2O_5)$) is expressed as
follows.
(Eq. 4) $k(N_2O_5) = 0.25 \times c(N_2O_5) \times S_a \times \gamma(N_2O_5)$
where $c(N_2O_5)$ represents the average molecular velocity of $N_2O_5$. The rate constants
($k_1$, $k_i$, and $k_{NO+NO_3}$) and equilibrium constant ($K_{eq}$) are calculated as temperature-
dependent parameters.

$\gamma(N_2O_5)$ and $\varphi(ClNO_2)$ were estimated using steady-state analysis in applicable cases
(Brown et al., 2006). This method assumes a steady state of $N_2O_5$, which means that





the production rate of $N_2O_5$ is equal to its loss rate. We adopted the criteria described
by Xia et al. (2020) to select the cases, namely low concentrations of NO, an increasing
trend of $ClNO_2$ concentrations, and stable air masses. Equation (5) was then established
by plotting $\tau(N_2O_5)^{-1} \times [NO_2] \times K_{eq}$ against $0.25 \times S_a \times C_{N_2O_5} \times [NO_2] \times K_{eq}$, with
$\gamma(N_2O_5)$ as the slope and $k(NO_3)$ as the intercept in the linear regression (Brown et al.,
2003). Here, the derived $\gamma(N_2O_5)$ was accepted when the regression had $R^2 > 0.5$ and
$k(NO_3) > 0$.
(Eq. 5) $\tau(N_2O_5)^{-1} \times K_{eq} \times [NO_2] \approx 0.25 \times C_{N_2O_5} \times S_a \times K_{eq} \times [NO_2] \times \gamma(N_2O_5) + k(NO_3)$
$\varphi(ClNO_2)$ was then calculated using the following equation:
(Eq. 6) $\varphi(ClNO_2) = \dfrac{d[ClNO_2]/dt}{k(N_2O_5)[N_2O_5]}$
where $d[ClNO_2]/dt$ and $[N_2O_5]$ represent the increasing rate of $ClNO_2$ production and
the average concentration of $N_2O_5$, respectively within the selected cases.

2.5 Box model
An observation-based chemical box model was utilized to simulate the
concentrations of Cl and $RO_x$ radicals and the production and loss pathways of $O_3$. The
detailed model description is available in Peng et al. (2020). Based on Master Chemical
Mechanism (MCM) v3.3.1 (Jenkin et al., 2015), Peng et al. (2020) modified the
chemical mechanisms to include up-to-date gas-phase chlorine and bromine chemistry.
The observed $N_2O_5$, $ClNO_2$, $NO_x$, HONO, $O_3$, $jNO_2$, and related species were
constrained in the model for every 10 minutes of model time, after interpolating or
averaging the data (Table S3). The mixing ratios of oxygenated volatile organic
compounds (OVOCs) and VOCs (Section 2.3) were constrained every hour. We
assumed the mixing ratio of $CH_4$ to be constant at 2000 ppbv (Tan et al., 2017). The
photolysis frequencies of $ClNO_2$, $O_3$, and other species were simulated according to the
solar zenith angle using the Tropospheric Ultraviolet and Visible (TUV) Radiation
model and scaled by the observed $jNO_2$ values. Numerical experiments were conducted
by constraining (Case 1) and not constraining $ClNO_2$ data (Case 2) at each site. The
differences in the radical concentrations and $O_3$ budgets between Cases 1 and 2
represented the effect of $ClNO_2$. For example, the increase in $RO_x$ (%) due to $ClNO_2$
was calculated by $(RO_x\_w - RO_x\_wo)/RO_x\_wo$, where $RO_x\_w$ represents the
concentration of $RO_x$ in Case 1 with $ClNO_2$ constrained in the model and $RO_x\_wo$
represents the concentration of $RO_x$ in Case 2 without $ClNO_2$ constrained.

**3. Results**
3.1 Overall measurements, diurnal patterns and comparison with other studies
The time series of $N_2O_5$ and $ClNO_2$ levels in the three campaigns are displayed in
Fig. 1. Overall, elevated levels of $N_2O_5$ and $ClNO_2$ were observed with different
patterns at each site. The ground sites (Wangdu and Beijing) were characterized by high
$NO_x$ levels (83.2 ± 81.3 ppbv and 35.6 ± 27.3 ppbv, respectively) and low $O_3$ levels
(8.5 ± 8.8 ppbv and 17.3 ± 11.4 ppbv, respectively), whereas the mountain site, Mt. Tai,
was marked by relatively lower $NO_x$ levels (2.4 ± 2.0 ppbv) and higher $O_3$ levels (64.6
± 14.7 ppbv) (Fig. S6). The campaign-averaged mixing ratios of $ClNO_2$ were similar at





the ground sites ($71 \pm 132$ pptv and $76 \pm 103$ pptv in Wangdu and Beijing, respectively),
and were significantly lower than that at Mt. Tai ($179 \pm 247$ pptv). The nocturnal ratio
of $ClNO_2/N_2O_5$ at each site displayed large day-to-day variability, which was positively
dependent on the ambient RH (Fig. S7) and, to a lesser extent, positively correlated
with $S_a$ (figure not shown).

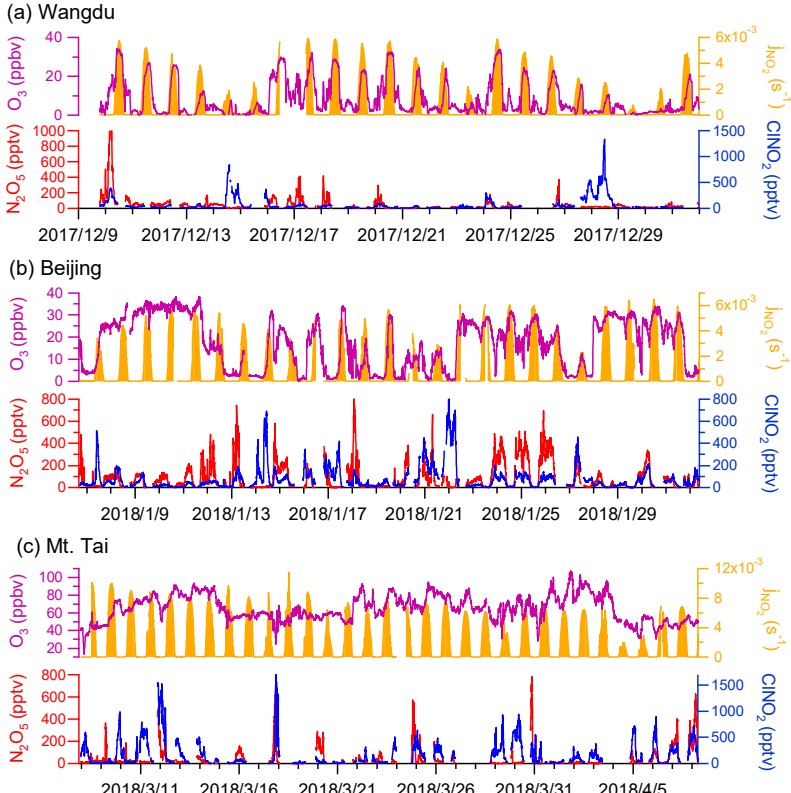

**Figure 1.** Overall observations of $N_2O_5$, $ClNO_2$ and related species at the **(a)** Wangdu,
**(b)** Beijing, and **(c)** Mt. Tai sites.

298       The campaign-averaged diurnal patterns of the mixing ratios of $N_2O_5$, $ClNO_2$, and
related species are depicted in Fig. 2. $ClNO_2$ levels typically exhibited a daily cycle,
peaking at night and decreasing during the day. The diurnal pattern of $ClNO_2$ at the
Wangdu site in winter was an exception, with elevated concentrations (10–90
percentiles) around midday (12:00–14:00 local time; LT), which resulted from a
noontime peak in $ClNO_2$ concentrations during a few days at Wangdu. The detailed
observation results from each site are separately introduced as follows.

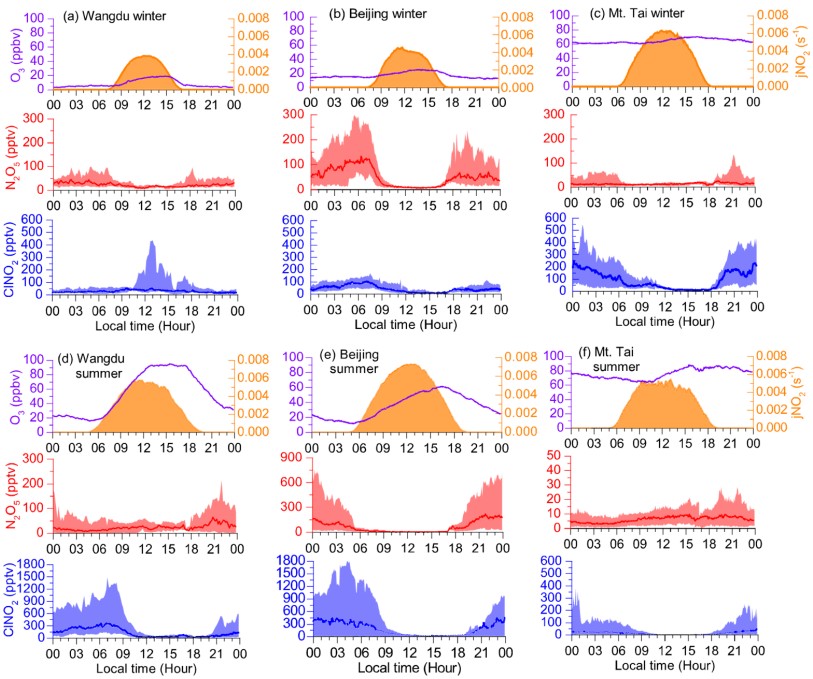

**Figure 2.** Diurnal average levels of $N_2O_5$, $ClNO_2$, $O_3$, and $jNO_2$ observed at the Wangdu, Beijing, and Mt. Tai sites throughout the campaign in winter (this study) and previous summer field studies (Table 1). The shaded areas indicate the 10th and 90th percentiles.

The nocturnal production of $ClNO_2$ was insignificant in Wangdu despite the presence of abundant $Cl^-$ ($3.3 \pm 3.2$ µg m$^{-3}$ throughout the observation), which likely originated from the intensive residential coal combustion in the area (Peng et al., 2020). The Wangdu site experienced high mass concentrations of $PM_{2.5}$ (a maximum of approximately 450 µg m$^{-3}$) and very large mixing ratios of NO (a maximum of approximately 350 ppbv). The wind rose analysis showed that the high concentrations of NO originated from the west of the sampling site where two major roads were located. Numerous heavy-duty trucks on these roads were responsible for high NO concentrations. The presence of abundant NO inhibited $N_2O_5$ formation by consuming $O_3$ and $NO_3$ at the Wangdu site. When the ambient concentrations of NO substantially decreased, e.g., on 10 December, the $N_2O_5$ mixing ratios increased to 1 ppbv. The mixing ratios of $ClNO_2$ were mostly low (< 200 pptv) during the night. However, significant daytime peaks in $ClNO_2$ mixing ratios were observed on 14 and 28 December, reaching approximately 0.8 ppbv and 1.3 ppbv, respectively. The daytime peaks in $ClNO_2$ concentrations at the three sites are discussed in detail in Section 3.3. For comparison, the ambient mixing ratios of NO in the summer campaign at Wangdu were much lower (mostly 0-10 ppbv) and $O_3$ mixing ratios were much higher (i.e., exceeded 90 ppbv on most days), which favored the production of $N_2O_5$ and $ClNO_2$ (Tham et al., 2016).



The winter Beijing observations showed that there was significant production of
$N_2O_5$ but limited conversion of $N_2O_5$ to $ClNO_2$ in dry conditions. The observation
period in Beijing was divided into polluted days (24-h $PM_{2.5} > 75$ µg m$^{-3}$; China's Grade
II air quality standard for $PM_{2.5}$) and clean days (24-h $PM_{2.5} < 35$ µg m$^{-3}$; Grade I
standard). The polluted periods were characterized by simultaneous high levels of $PM_{2.5}$
and NO, e.g., on 19 January. The clean periods were marked by relatively high mixing
ratios of $O_3$, low levels of $PM_{2.5}$ and $NO_x$, e.g., from 8 to 11 January. Both polluted and
clean conditions were unfavorable for $ClNO_2$ formation owing to the high
concentrations of NO on the polluted days and the low concentrations of $NO_2$ and
aerosols on the clean days. Moreover, the RH observed in Beijing was typically below
40 %, which indicated relatively slow heterogeneous loss of $N_2O_5$ and slow $ClNO_2$
formation. Consequently, $N_2O_5$ mixing ratios frequently accumulated to elevated levels,
exceeding 0.4 ppbv on 10 of the 26 observation nights, and the mixing ratio of $ClNO_2$
was mostly below 0.4 ppbv. The highest mixing ratios of $ClNO_2$ were observed (up to
approximately 0.8 ppbv) when the site occasionally intercepted air masses with a higher
RH (approximately 75 %), e.g., on the night of 22 January. This result is similar to the
previous observation in Beijing (Xia et al., 2019), in which the ratio of $ClNO_2$ to $N_2O_5$
increased significantly from late spring with a low RH (10–30 %) to early summer with
a higher RH (20–80 %). The overall mixing ratios of $ClNO_2$ in the present Beijing study
in winter were also significantly lower than those reported in summer (maximum of 1.4
ppbv to 2.9 ppbv) in other studies (Breton et al., 2018; Zhou et al., 2018).
Elevated mixing ratios of $ClNO_2$ (i.e., above 0.5 ppbv) were frequently recorded at
the Mt. Tai station. High concentrations of $PM_{2.5}$ ($34.5 \pm 27.3$ µg m$^{-3}$) and high RH
($63.6 \pm 27.1$ %) favored the $ClNO_2$ formation at Mt. Tai. The maximum level of $ClNO_2$
(approximately 1.7 ppbv) was observed just before midnight on 18 March, which was
slightly lower than the highest concentration observed at Mt. Tai in the summer of 2014
(Wang et al., 2017c). The elevated concentrations of $ClNO_2$ observed in the previous
summer study at Mt. Tai were due to emissions from distinct coal-fired power plants,
whereas this winter study found that coal burning had less effect on concentrations of
$ClNO_2$. The campaign-averaged levels of $SO_2$ and particulate $SO_4^{2-}$ were $1.6 \pm 1.6$ ppbv
and $3.6 \pm 2.9$ µg m$^{-3}$, respectively, during the winter observations, which were
significantly lower than those observed in the summer campaigns ($2.9 \pm 3.7$ ppbv and
$14.8 \pm 9.0$ µg m$^{-3}$, respectively). The reduced effect from coal-fired power generation
was due to the continued decrease in $SO_2$ emissions during 2014-2018 and less transport
of emissions from the ground to the Mt Tai site (1534 m a.s.l.) in late winter and early
spring compared with that in summer.
We compared the observed winter concentrations of $ClNO_2$ with those reported in
previous studies in Asia, North America, and Europe (Fig. 3). The highest winter
concentrations of $ClNO_2$ to date were observed in southern China, with a maximum
level of 4.7 ppbv at a mountain top in Hong Kong in aged urban/industrial plumes from
the Pearl River Delta (PRD) (Wang et al., 2016) and 8.3 ppbv during a severe pollution
episode within the PRD (Yun et al., 2018). The high-concentration $ClNO_2$ events in



southern China were due to concurrent high levels of PM$_{2.5}$ and O$_3$ (e.g., 400 μg m$^{-3}$
and 160 ppbv found by Yun et al., 2018), which contrasts the high concentrations of
PM$_{2.5}$ and low concentrations of O$_3$ over northern China during the cold winter. The
winter mixing ratios of ClNO$_2$ in the US and Europe range from approximately 0.3 ppbv
in urban California (Mielke et al., 2016) and urban Manchester (Priestley et al., 2018),
respectively, to 1.3 ppbv in the outflow of coastal urban areas (Riedel et al., 2013;
Haskins et al., 2019). In general, the winter concentrations of ClNO$_2$ over northern
China were comparable to or slightly higher than those observed in the US and Europe.

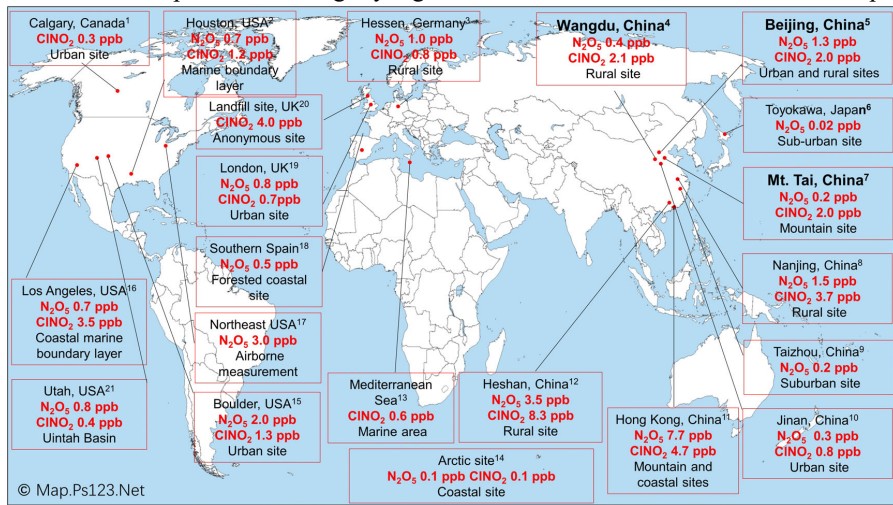


Figure 3. Previous observations of ClNO$_2$ and N$_2$O$_5$ levels worldwide. Observation
sites in this study are shown in bold. The ClNO$_2$ and N$_2$O$_5$ levels shown are the highest
that were measured at these sites. Footnotes associated with the locations refer to the
references as follows. 1. (Mielke et al., 2011; Mielke et al., 2016). 2. (Osthoff et al.,
2008). 3. (Phillips et al., 2012). 4. (Tham et al., 2016; Liu et al., 2017). 5. (Wang et al.,
2017a; Breton et al., 2018; Wang et al., 2018; Zhou et al., 2018; Xia et al., 2019). 6.
(Nakayama et al., 2008). 7. (Wang et al., 2017c). 8. (Xia et al., 2020). 9. (Wang et al.,
2019a). 10. (Wang et al., 2017b). 11. (Wang et al., 2016; Yun et al., 2017; Yan et al.,
2019). 12. (Yun et al., 2018). 13. (Eger et al., 2019). 14. (Apodaca et al., 2008;
McNamara et al., 2019). 15. (Thornton et al., 2010; Riedel et al., 2013). 16. (Riedel et
al., 2012; Mielke et al., 2013). 17. (Brown et al., 2006; Brown et al., 2007). 18.
(Crowley et al., 2011). 19. (Bannan et al., 2015). 20. (Bannan et al., 2019). 21. (Edwards
et al., 2013; Wild et al., 2016).

3.2 NO$_3$ production and loss pathways
To gain insight into the processes controlling the variability in concentrations of
ClNO$_2$, nocturnal $P$(NO$_3$) and NO$_3$ loss pathways were compared using Eqs. (1-5) in
Section 2.4. The average $P$(NO$_3$) was comparable at the three sites in winter, ranging
from 0.15 ppbv h$^{-1}$ to 0.25 ppbv h$^{-1}$, and these rates were significantly lower than the
respective summer values (Fig. 4a). The lower $P$(NO$_3$) in winter was caused by both
lower k$_1$ and lower [NO$_2$] × [O$_3$] in winter (see Eq. 1). Nighttime NO$_3$ removal through
NO$_3$ and N$_2$O$_5$ was estimated by comparing $k$(NO$_3$) × [NO$_3$] (Eqs. 2–3) and $k$(N$_2$O$_5$) ×
[N$_2$O$_5$] (Eqs. 4–5). The average γ(N$_2$O$_5$) values derived from each campaign (Table S4
and Fig. S8) were used in Eq. (4). The nighttime NO$_3$ loss via NO titration and VOC
oxidation was greater than the N$_2$O$_5$ heterogeneous loss in all the winter and summer
campaigns (Fig. 4b). These were the campaign average results. In contrast, the N$_2$O$_5$
loss was greater than the NO$_3$ loss in selected cases in summer at Mt. Tai (Wang et al.,
2017c). To determine the nocturnal loss of NO$_3$, we further compared the N$_2$O$_5$/NO$_3$
ratio and γ(N$_2$O$_5$) at the three sites.


**Figure 4**. Comparison of $P$(NO$_3$) and loss pathways of NO$_3$ during the winter and
summer observations over the NCP. W and S are abbreviations for winter and summer,
respectively.

The thermal decomposition of N$_2$O$_5$ was suppressed in winter and resulted in high
ratios of N$_2$O$_5$/NO$_3$ (Fig. 5a; up to approximately 1000), which favored N$_2$O$_5$ loss over
NO$_3$ loss. However, the γ(N$_2$O$_5$) in winter was systematically lower than that in summer
(Fig. 5b), which indicated slower N$_2$O$_5$ loss in winter. This result differs from previous
laboratory studies, which reported larger γ(N$_2$O$_5$) on (NH$_4$)$_2$SO$_4$ aerosols at lower
temperatures (Hallquist et al., 2003; Griffiths and Anthony Cox, 2009). It is possible



that other factors, such as RH and aerosol composition (aside from $(NH_4)_2SO_4$), had a
large influence on $\gamma(N_2O_5)$. The limited number (2–4) of $\gamma(N_2O_5)$ values obtained in
each winter campaign (Table S4) may have also caused a bias in the estimation of the
overall $\gamma(N_2O_5)$. The opposite effects – a higher $N_2O_5/NO_3$ ratio and lower $\gamma(N_2O_5)$ in
winter – offset each other in Wangdu (Fig. 4b) but favored $N_2O_5$ loss in Beijing and
$NO_3$ loss at Mt. Tai compared with those in the respective summer campaigns. The
higher concentrations of $ClNO_2$ at Mt. Tai during the winter campaigns may be
attributable to higher $\varphi(ClNO_2)$ values in Mt. Tai (Fig. 5c).

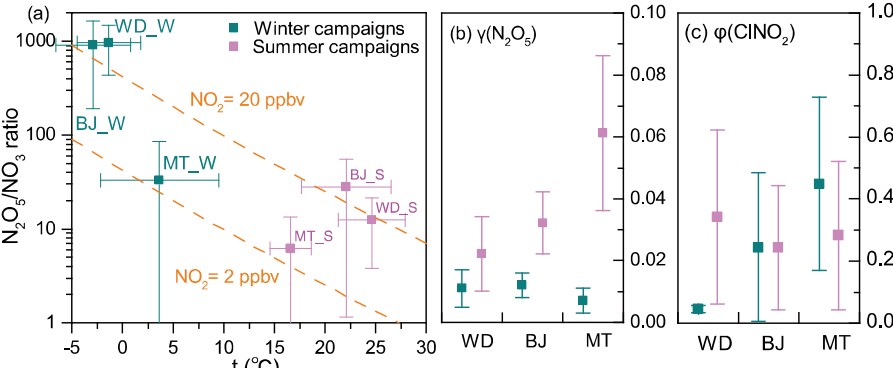

**Figure 5**. Comparison of the **(a)** $N_2O_5/NO_3$ ratio, **(b)** $\gamma(N_2O_5)$, and **(c)** $\varphi(ClNO_2)$ during
the winter and summer campaigns. Square dots and error bars indicate the average
values and standard deviations, respectively.

3.3 Daytime peaks in $ClNO_2$ concentrations
In the winter campaigns, high concentrations of $ClNO_2$ were sustained after sunrise.
Distinct peaks in $ClNO_2$ concentrations were observed on 3–4 days in each campaign,
as shown in Fig. 6. Other daytime cases from the three sites are shown in Fig. S9–11.
The validity of the daytime peaks was checked by performing isotopic analysis of
$ClNO_2$, background detection, and onsite calibration. The signals of $I^{35}ClNO_2^-$ and
$I^{37}ClNO_2^-$ were well correlated ($R^2 > 0.99$) during daytime peaks in $ClNO_2$
concentrations (Fig. S3a–c) and calibrations (Fig. S3d–f). The ratio of $I^{37}ClNO_2^-$ to
$I^{35}ClNO_2^-$ (0.32–0.35) was consistent with the natural isotopic ratio of $^{37}Cl$ to $^{35}Cl$. The
background signals of $ClNO_2$ were checked when its daytime peaks in concentrations
were observed, and no increase in the background was found. These results confirmed
that the daytime peaks in $ClNO_2$ concentrations were real atmospheric phenomena.

The daytime-$ClNO_2$ episodes usually occurred from 10:00 to 11:00 LT at each site.
The highest daytime mixing ratio of $ClNO_2$ was 1.3 ppbv (5-minute average) observed
at 11:30 on 28 December 2017 in Wangdu. In comparison, the daytime $ClNO_2$
concentration observed in the previous summer study at Wangdu (Tham et al., 2016)
reached a maximum in the early morning (08:00 LT) and declined to several pptv at
11:00 am. Attenuated solar radiation was observed during the days with daytime peaks





in ClNO$_2$ concentrations. For example, the daily maximum rates of $j$NO$_2$ (1-minute
average) for the Wangdu case shown in Fig. 6a (2.5 × 10$^{-3}$ s$^{-1}$) was significantly lower
than the highest rate observed during this campaign (6.0 × 10$^{-3}$ s$^{-1}$). The attenuated solar
radiation reduced the photolysis of ClNO$_2$, which allowed it to persist for a longer
period during the day. The chemical data showed contrasting features during the
daytime peaks in ClNO$_2$ concentrations at the three sites. At Wangdu, ClNO$_2$
concentrations showed a sharp peak while the concentrations of other pollutants
decreased (Fig. 6a); in Beijing, the daytime peak in ClNO$_2$ concentrations appeared
with little simultaneous change in the NO$_3^-$, NO$_x$, and O$_3$ levels after sunrise (Fig. S10a).
In two cases, daytime peaks of ClNO$_2$ concentrations at Mt. Tai (Fig. 6c and Fig. S11c)
occurred together with significant increases in NO$_3^-$, NO$_x$, and PM$_{2.5}$ levels, whereas
O$_3$ concentrations decreased after sunrise and resumed its previous levels.

469        The daytime peaks in ClNO$_2$ concentrations were likely caused by the transport of
air masses to the respective sites. In situ production of ClNO$_2$ was limited during the
days on which significant daytime ClNO$_2$ occurred, because the mixing ratios of N$_2$O$_5$
were near the detection limit of the instrument (several pptv). The photochemical
lifetime of ClNO$_2$ at 10:00 am LT was estimated to be 1–2 h, based on the inverse of
$j$ClNO$_2$, which allowed the transport of ClNO$_2$ produced elsewhere to the observation
sites. As daytime peaks in ClNO$_2$ concentrations appeared at both the ground and
mountain sites, the high-ClNO$_2$ region may exist in the residual layer above the
nocturnal mixing layers. At sunrise, ClNO$_2$-rich air masses may be transported
downward to the ground sites (Wangdu and Beijing) and upward to the mountain-top
site (Mt. Tai). The downward transport of ClNO$_2$ at Wangdu in summer has been
illustrated by Tham et al. (2016), and the upward transport to the top of Mt. Tai has also
been implicated by the increasing daytime concentrations of O$_3$ and other pollutants
(e.g., Gao et al., 2005; Zhou et al., 2009; Jiang et al., 2020). Measurements in the
residual layers are needed to further investigate the transport of ClNO$_2$ within the entire
boundary layer.

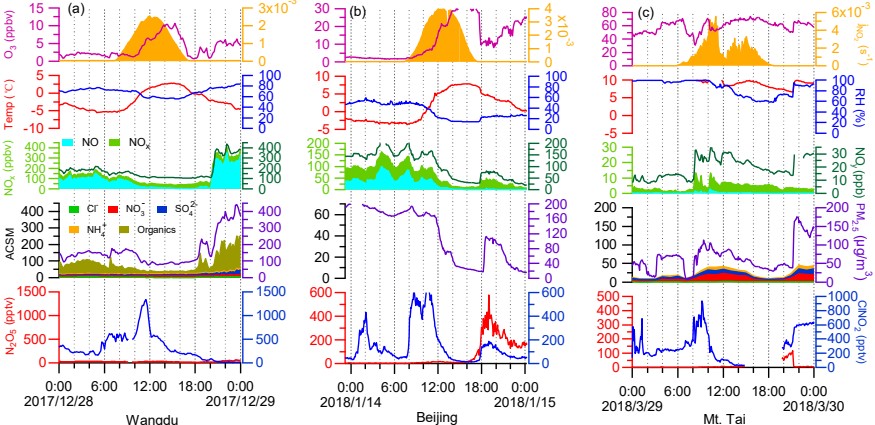

**Figure 6.** Examples of daytime peaks of ClNO$_2$ levels observed at **(a)** Wangdu, **(b)**
Beijing, and **(c)** Mt. Tai in the winter campaigns. These examples show the highest





levels of daytime ClNO$_2$ at each site. The ionic composition of aerosols was not
available on 14 January 2018, owing to an instrument problem.
3.4 Impact of daytime ClNO$_2$ on atmospheric oxidation capacity
We used the box model (Section 2.5) to show the impact of ClNO$_2$ on photochemical
oxidation at the three sites (Fig. 6a–c). In campaign-averaged conditions, the impact of
ClNO$_2$ was minor, owing to the low daytime concentrations of ClNO$_2$. The daytime-
averaged $P$(Cl) (06:00–18:00 LT) from ClNO$_2$ photolysis was in the range of 0.03–0.06
ppbv h$^{-1}$, with the peak values of 0.07–0.12 ppbv h$^{-1}$, and the photolysis of ClNO$_2$
enhanced the daytime RO$_x$ concentrations by 1.3–3.8 % and net O$_3$ production by 1.3–
6.2 % at the three sites (figures not shown). Such impacts were lower than those during
summer at Wangdu (Tham et al., 2016).
However, the impact of ClNO$_2$ increased considerably in the cases of daytime-peak
concentrations, as shown in Fig. 7. The daytime-averaged $P$(Cl) values from ClNO$_2$
photolysis were 0.15 ± 0.13 (maximum of 0.46), 0.11 ± 0.09 (maximum of 0.32), and
0.19 ± 0.20 (maximum of 0.74) ppbv h$^{-1}$ at Wangdu, Beijing, and Mt. Tai, respectively
(Fig. 7a–c). The winter $P$(Cl) peak in Wangdu (Fig. 7a, 0.46 ppbv h$^{-1}$) was twice the
summer average value (0.24 ppbv h$^{-1}$) (Tham et al., 2016). $P$(Cl) from other sources
(e.g., the HCl + OH reaction) was minor (8.8–14.5 %) during these cases. The relative
importance of ClNO$_2$ in primary radical production varied among these sites. ClNO$_2$
had a minor contribution in Beijing but became increasingly important in Wangdu and
Mt. Tai (Fig. 7b, c). HONO photolysis was the most important source of OH at the two
ground sites, whereas O$_3$ was also important at Mt. Tai.
The liberated Cl (mostly from ClNO$_2$ photolysis) accounted for 28.5–57.7 % of the
daytime (06:00–18:00 LT) oxidation of alkanes, 6.1–13.7 % of that of alkenes, 5.3–
14.2 % of that of aromatics, and 4.6–6.0 % of that of aldehydes in the cases of high
levels of daytime ClNO$_2$. The Cl + VOCs reactions enhanced the production of OH,
HO$_2$, and RO$_2$ by up to 15–22 %, 24–31 %, and 36–52 %, respectively (Fig. 7d–f). The
photolysis of ClNO$_2$ increased the daytime net O$_3$ production by 5.4 ppbv (18 %), 2.8
ppbv (17 %), and 2.6 ppbv (13%) at Wangdu, Beijing, and Mt. Tai, respectively (Fig.
7g–i). These results indicate the considerable impact of daytime ClNO$_2$ on the
atmospheric oxidative capacity and production of secondary pollutants.
The impact of Cl in the NCP is likely larger than the result shown above. Our model
calculations considered photolysis of ClNO$_2$ (and HCl + OH) as the source of Cl, but
not other photolabile Cl-containing gases. However, in the Wangdu field campaign, we
frequently observed elevated daytime concentrations of bromine chloride (BrCl) and
molecular chlorine (Cl$_2$), which dominated the Cl production (Peng et al., 2020). In
addition, our ClNO$_2$ measurements were conducted at polluted ground-level sites and
at a high mountain site (1534 m a.s.l.), which are not in the nocturnal residual layer
where strong ClNO$_2$ production is expected to occur (Zhang et al., 2017). It is thus
highly desirable to measure ClNO$_2$ in the residual layer in future studies to

comprehensively assess the role of $ClNO_2$ in the lower part of the atmosphere.

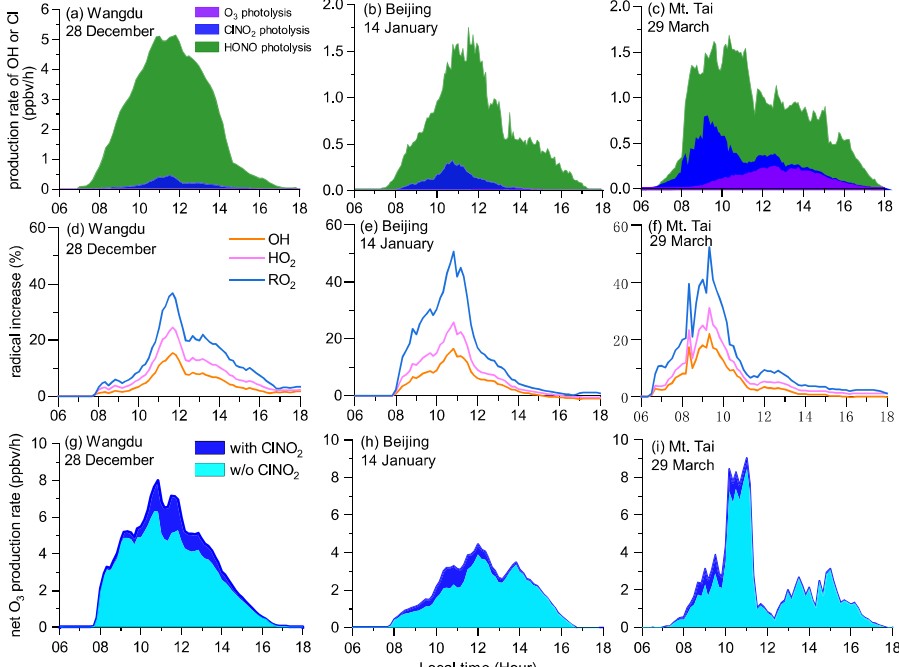

**Figure 7.** The impact of $ClNO_2$ photolysis on atmospheric oxidation during daytime-$ClNO_2$ episodes: **(a)** primary radical production from the photolysis of $O_3$, $ClNO_2$, and HONO; **(b)** percentage increase in OH, $HO_2$, and $RO_2$ due to $ClNO_2$ photolysis (Section 2.5); and **(c)** enhancement of net $O_3$ production rates due to $ClNO_2$ photolysis.

**4. Summary and conclusions**

Observations of $ClNO_2$ and related species were conducted at urban, rural, and mountain-top sites in the winter of 2017–2018 in the NCP, which suffers from severe winter haze pollution. The winter measurements showed lower concentrations of $ClNO_2$ compared with those in previous summer observations. The campaign averaged $NO_3$ loss at night dominated over the $N_2O_5$ loss at all the sites due to high NO concentrations, and in situ $ClNO_2$ formation was generally insignificant. However, high levels of daytime $ClNO_2$ (exceeding 1 ppbv) were observed at the three sites. We suggest that $ClNO_2$ was efficiently produced in the nocturnal residual layer and was transported to ground-level and high-elevation sites. The daytime concentrations of $ClNO_2$ had great effects on the production of Cl, $RO_x$, and $O_3$. Vertical measurements of the concentrations of $ClNO_2$ and related compounds are needed to better understand the distribution and impact of these species in the lower part of the troposphere.

*Data availability.*

The datasets described in this study is available by contacting the corresponding author (cetwang@polyu.edu.hk).





*Author contributions.*

TW designed this study. JC, YM, LX, JG, and HL provided field measurement sites. MX, XP, and WW conducted the CIMS measurements. CY, ZW, YJT, HC, CZ, PL, and XW provided supporting data. XP and WW performed the box model simulation. MX analyzed and virtualized the research data. MX and TW wrote the manuscript with discussions and comments from all co-authors.

*Competing interests.*

The authors declare that they have no conflict of interest.

*Acknowledgments.*

The authors are grateful to Yujie Zhang, Fang Bi, Zhenhai Wu, and Xi Cheng for providing supporting data in Beijing. The authors acknowledge helpful discussions and opinions from Peng Wang, Xiao Fu, logistics support from Liwei Guan in Wangdu, and the meteorological observatory at Mt. Tai for providing experiment platforms.

*Financial support.*

This work was funded by National Natural Science Foundation of China (grant nos. 91544213, 91844301, and 41922051), the Hong Kong Research Grants Council (grant nos. T24-504/17-N and A-PolyU502/16), and National Key Research and Development Program of China (grant no. 2016YFC0200500).

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
