# Peer review of "Winter observations of ClNO2 in northern China: Spatiotemporal variability and"

_Atmospheric Chemistry and Physics, 2021_

## Referee Comment (RC1)

**Winter observations of ClNO2 in northern China: Spatiotemporal variability and insights into daytime peaks**

Men Xia1 , Xiang Peng1 , Weihao Wang1,8, Chuan Yu1,2, Zhe Wang6 , Yee Jun Tham7 3 , Jianmin Chen4 , Hui Chen4 , Yujing Mu5 , Chenglong Zhang5 , Pengfei Liu5 , Likun Xue2 4 , Xinfeng Wang2 , Jian Gao3 , Hong Li3 , and Tao Wang1

**General Comments:**

This paper compares the formation of ClNO2 and its impact on the tropospheric radical budget at 3 ground sites in China during winter and summer over an extended measurement period. It is important in that it shows in places subject to fresh emissions of NO, that ClNO2 formation can be even more important in the summer and during the daytime when compared to its formation during the winter in the same places.

The body of the paper details that (1) less photochemical production of O3, (2) more fresh NO emissions at the Wangdu & Beijing sites in winter and (3) especially dry conditions at the Beijing site in winter are responsible for suppressing the NO3 radical production // dominating the loss of NO3 & therefore suppressing ClNO2 production in winter when compared to summer. This, in addition to seasonal differences in their calculated uptake coefficients of N2O5 and yields of ClNO2 ultimately explain the lower concentrations of ClNO2 during winter compared to summer at the sites. The observations and analysis presented highlight that ClNO2 can be important during summer and during the day, and that the behavior of observed ClNO2 is explainable by our understanding of its chemistry under different conditions (e.g. more NO, less O3, low RH). My biggest concern with the paper in its current form is that the abstract and conclusion focus largely on the fact that "observed ClNO2 is higher in summer than winter at these sites" and the underlying messages of "why this occurs is in line with the current understanding of the formation of ClNO2"  and "the summer/winter trends at these ground sites which experience a lot of fresh pollution are not generally representative of trends we expect in the residual boundary layer where ClNO2 formation is higher" may be lost on the casual reader. I would like to see the abstract and conclusions revised to better communicate that portion of the results (which is well communicated in the body of the text).

Overall, the methods and assumptions are well outlined, the discussion is detailed, the results are presented in a logical structure, the language is clear (easy to read- well done!), and their conclusions are well reasoned. The analysis within represents a clear step forward in our understanding of the formation and role of ClNO2 in the troposphere under different conditions. There are a few important citations missing from the paper, that I believe should be added, and I have suggested several modifications to figures that could improve the overall communication of the results. Ultimately, I recommend that this paper be accepted with minor revisions.

**Specific Comments:**

**Title:** It's probably worth mentioning the summer /winter comparison which makes this work novel or the control of the NO emissions in the title. Maybe "Local seasonal emissions control ClNO2 formation in northern China: Spatiotemporal variability and insights in into daytime peaks" or "Comparing the sensitivity of winter and summer ClNO2 formation in Northern China to local emissions: Spatiotemporal variability and insights in into daytime peaks"?

**Line 57-59:** I suggest adding a sentence about the impact of Cl radicals in the non-polluted troposphere (since that is much of their impact on a global scale). This will set up your readers to better interpret the differences you see between sites since they experience more fresh pollution than studies in other regions.

**Line 70-71:** I would also cite Simpson et al., 2015 (ACS: Tropospheric Halogen Chemistry: Sources, Cycling, and Impacts https://pubs.acs.org/doi/10.1021/cr5006638) either here or somewhere else in the introduction.

**Line 98-101:** This statement is not an accurate summation of the reference since they conclude that Cl *is* the dominant radical source in the polluted MBL there (at least in the early morning). I would suggest revising it to: "The role of ClNO2 in the radical budget could be more important than that of OH in winter, because OH production is reduced in winter owing to lower concentrations of O3 and H2O vapor in this season. Haskins et al., 2019 recently confirmed that, even when compared to OH, Cl atoms from ClNO2 photolysis can be the dominant early morning radical source and the dominant integrated daily radical source over the polluted marine boundary layer downwind of the northeast US."

**Lines 109-119:** I would also mention longer NOx lifetimes allows NOx to spread further distances from its local sources during winter. You may also clarify that the variability in seasonal Cl availability you reference is unique to the NCP.

**Line 118-119:** This line is quite a strong statement. I suggest reframing it, particularly since the sites examined in this work are subject to quite substantial changes in the precursor conditions. (e.g. "Because of the competing trends and variability in chemical precursors to N2O5 and ClNO2, it is not clear whether ClNO2 formation is always more prevalent during winter compared to summer, particularly in regions that experience large variability in the conditions of the advected air masses they experience.")

At some point in the introduction, there needs to be some discussion about our understanding of ClNO2 formation in regions subject to fresh pollution (e.g. lots of fresh NO) verses in aged polluted air masses verses in clean air. All the results in the paper can be explained with what we already know about ClNO2 formation in different types of air masses- but you need to set the reader up for what to expect in each different type of air mass before you get to the results. I think it will set up the discussions of why the trends at the Wangdu site look as they do in a way that will be useful to readers less familiar with the formation process of ClNO2.

**Table 1:** I think the "Site Categories" are misleading/unclear. The discussion of each site is excellent & builds a great picture of the conditions experienced, but this table does not summarize those well. On line 158-159 you state that the Wangdu site experiences heavy pollution from coal burning and road traffic. In the results section (lines 285-288) you state that both the Wangdu and Beijing sites experience

high NOx and low O3. However, Wangdu is categorized as a "rural" site, while Beijing is categorized as an "urban" site. While the Wangdu site is certainly more remote than the upwind Beijing site, the category of "rural" is typically used to describe low CO/NOx conditions and urban used to describe places with more CO/ NOx. I suggest either recategorizing Wangdu as "remote polluted" and Beijing as "upwind urban polluted" or adding a column to the table with the average daily NOx and O3 concentrations observed in each observation period. I think the latter might be more useful (because then you could tell that the sites experience rather different pollution conditions in the different seasons). Finally categorizing Mt. Tai as a "mountain" site seems redundant and non-descriptive- perhaps a "remote residual layer" site or "remote clean" site. The abstract and conclusion where these categorical descriptors are used should also be updated (lines 25, 541)

**Lines 224-244:** A citation for the rate constants used in the box model for each of these equations is needed for reproducibility.

**Line 230-257:** Some discussion about k(NO3) is needed. What VOCs were used? If you did not measure the full suite of VOCs then you would underestimate k(NO3) in Eq. 2 thereby impacting the calculated uptake of N2O5 and yield of ClNO2. A statement about the uncertainty arising from this is needed at minimum.

**Figure 1**: This figure could be improved by making the various y-axis limits consistent across all sites when possible. I see no reason why the jNO2 axis can't be consistent across all panels with a max value at 8*10-3 (as it is on later figures). I would also label the site not with just their name but their "category" e.g. Wangdu: remote polluted). If visibility of the data is impacted, I suggest highlighting the axis differences at minimum.

**Figure 2:** Again, this figure could be improved if the y-axis limits were consistent when possible, allowing for easier comparisons. I also think it would be interesting to see the winter and summer data on the same plot rather than split up as it is. Perhaps by using color to denote winter verses summer rather than chemical species/ axis and a translucent shading.

**Lines 310-350:** There is a really important discussion here about the role that shifting NO concentrations plays in driving the difference in the winter and summer measurements at Wangdu and Beijing. However, this important information is not currently communicated in any of the figures of the paper, but easily could be given the observations made.

> *(This may be beyond the scope of the authors at this state in revision… But I think it would be interesting to see how ClNO2 formation compared in the winter vs summer observations but grouped by daily peak NO (or NOx) concentrations or daily averaged NO (or NOx) conditions during the day and during the nighttime. (e.g. a bar chart showing ClNO2 concentrations on the y axis, grouped by NO concentrations on the x axis with a bar for winter right next to a bar for summer in each of the NO groupings along the x axis. If the winter/summer average diurnal profiles were combined on Figure 2 in to only 3 panels, you could then show such a figure in panel d, e, & f for each site. It would show how the distribution of NOx as NO vs. NO2 changed between seasons, as well as likely organize why you see more ClNO2 when there is less NO, but more NO2… If that didn't organize well, even showing ClNO2 concentrations grouped by the calculated rate of production of NO3 would communicate the main message of the paper while explaining the average diurnal profiles in a visual way. )*

**Figure 3:** This would be a great summary figure to have in the literature. However, there appear to be several missing measurements of ClNO2/N2O5, particularly those over the US. The ones off the top of my head which are missing are as follows, but I would encourage the authors to ensure they have included all measurements to date, as I expect there at least a few other observations that are missing…

Faxon et al., 2015: https://www.mdpi.com/2073-4433/6/10/1487

Haskins et al., (2018): https://agupubs.onlinelibrary.wiley.com/doi/full/10.1029/2018JD028786

McDuffie et al., 2019: https://acp.copernicus.org/articles/19/9287/2019/

Jeong et al., (2019) : https://acp.copernicus.org/articles/19/12779/2019/

**Line 401-408:** These are potentially the most important results of the paper and I'd like to see the few sentences describing them be communicated more clearly. E.g. Explicitly explain why k1 is lower (wintertime temperatures, etc). Explicitly explain why you have lower [NO2] *[O3] in winter (e.g. less photochemical production of O3, more NO in winter titrating available O3, despite longer NO2 lifetimes in winter). Also while you give avg NOx conditions of the sites during winter there is no discussion about what those are during summer at these sites. Adding that info to Table 1 as previously suggested and perhaps in the paragraphs where the site results are individually introduced would set this discussion up better.

**Figure 4:** Similar to my comments about Figure 2, I wonder if it would be useful to also show on panel A, not just the mean in the winter verses in the summer, but also with the winter and summer measurements at each site separated into two different NOx (or NO) regimes (e.g. fresh pollution present verses not? Also, on panel B would it be possible to show the fraction of loss from NO verses VOCs with hatching? In the text it should also be stated that the loss to VOCs is potentially underestimated (See prior comments).

**Lines 439-449:** You do not mention if you looked at how IH2O- was changing during these daytime peaks? I suspect its not a problem given the infield calibrations, but it is worth adding a sentence to mention you'd check that as well.

**Figure 7:** Why is HCHO photolysis not included? I expect it to be a large contributor to the radical budget in at least some of these sites in these periods… Additionally, there's evidence that the presence of Cl radicals oxidizing VOCs enhances the production of HCHO and therefore OH concentrations so the differences between the with and without ClNO2 cases would be underestimated without considering the contributions from HCHO as well? At least some discussion is needed as to why HCHO is not considered part of the radical budget of OH.

**Lines 514 –522:** I'd like to see these results compared to results from other papers that have done this sort of analysis (e.g. Riedel et al., 2012, Young et al., 2014, Haskins et al., 2019, etc.). The conditions of the sites analyzed in this work are sufficiently different from other in the literature (e.g. more fresh NO more coal burning, more aerosol SA, way more HONO) that contrasting your results to those paper's results provides novel insights into the variability of ClNO2 production/ importance to the radical budget during winter in different chemical regimes. Also, Young et al., 2014 (https://acp.copernicus.org/articles/14/3427/2014/ ) is relevant to this work & I'd suggest adding it as a citation.

**Lines 529-531:** This is the only time that the measurements presented in this paper are put into context into how much they matter in the context of the globe. I'd like to see this sentiment present in the conclusions/ abstract as well to contextualize the results. I suggest adding a statement like this to the conclusions between lines 546-547.

**Lines 541:** Again, I don't think its appropriate to categorize the sites as "rural" given their polluted conditions (change to "remote polluted"). Update this line to reflect changes in its "categorization" in Table 1 and in the Abstract.

**Technical Corrections**

**Line 84:** Bertram et al., 2009 should be cited as Bertram & Thornton 2009 as it only has 2 authors.

**Line 103:** This reference should be a citation to Haskins et al., 2018 (Wintertime Gas-Particle Partitioning and Speciation of Inorganic Chlorine in the Lower Troposphere Over the Northeast United States and Coastal Ocean https://agupubs.onlinelibrary.wiley.com/doi/full/10.1029/2018JD028786 ) rather than Haskins et al., 2019 (Anthropogenic control over wintertime oxidation of atmospheric pollutants https://agupubs.onlinelibrary.wiley.com/doi/10.1029/2019GL085498 )

**Line 142:** "mostly during the *heading* period" should likely be "mostly during the *heating* period" …

---

## Author Response (AR1)

**General Comments:**

This paper compares the formation of ClNO2 and its impact on the tropospheric radical budget at 3 ground sites in China during winter and summer over an extended measurement period. It is important in that it shows in places subject to fresh emissions of NO, that ClNO2 formation can be even more important in the summer and during the daytime when compared to its formation during the winter in the same places.

The body of the paper details that (1) less photochemical production of O3, (2) more fresh NO emissions at the Wangdu & Beijing sites in winter and (3) especially dry conditions at the Beijing site in winter are responsible for suppressing the NO3 radical production // dominating the loss of NO3 & therefore suppressing ClNO2 production in winter when compared to summer. This, in addition to seasonal differences in their calculated uptake coefficients of N2O5 and yields of ClNO2 ultimately explain the lower concentrations of ClNO2 during winter compared to summer at the sites. The observations and analysis presented highlight that ClNO2 can be important during summer and during the day, and that the behavior of observed ClNO2 is explainable by our understanding of its chemistry under different conditions (e.g. more NO, less O3, low RH). My biggest concern with the paper in its current form is that the abstract and conclusion focus largely on the fact that "observed ClNO2 is higher in summer than winter at these sites" and the underlying messages of "why this occurs is in line with the current understanding of the formation of ClNO2" and "the summer/winter trends at these ground sites which experience a lot of fresh pollution are not generally representative of trends we expect in the residual boundary layer where ClNO2 formation is higher" may be lost on the casual reader. I would like to see the abstract and conclusions revised to better communicate that portion of the results (which is well communicated in the body of the text).

Overall, the methods and assumptions are well outlined, the discussion is detailed, the results are presented in a logical structure, the language is clear (easy to read- well done!), and their conclusions are well reasoned. The analysis within represents a clear step forward in our understanding of the formation and role of ClNO2 in the troposphere under different conditions. There are a few important citations missing from the paper, that I believe should be added, and I have suggested several modifications to figures that could improve the overall communication of the results. Ultimately, I recommend that this paper be accepted with minor revisions.

Response: We appreciate the reviewer for the positive comments and helpful suggestions. Below is the response to each comment. The reviewers' comments are shown in black font followed by our responses and changes in the manuscript shown in blue and red, respectively. The corrections are also marked as red color in the revised manuscript. Please note that the line numbers mentioned below refer to the original submission (line numbers in the revised version have changed).

**Specific Comments:**

**Title:** It's probably worth mentioning the summer /winter comparison which makes this work novel or the control of the NO emissions in the title. Maybe "Local seasonal emissions control ClNO2 formation in northern China: Spatiotemporal variability and insights in into daytime peaks" or "Comparing the sensitivity of winter and summer ClNO2 formation in Northern China to local emissions: Spatiotemporal variability and insights in into daytime peaks"?

Response: We appreciate the reviewer to suggest new titles and have changed the title in the revised manuscript.

Changes in the manuscript:

Line 1-2: Winter $ClNO_2$ formation in the region of fresh anthropogenic emissions: Seasonal variability and insights into daytime peaks in northern China

**Line 57-59:** I suggest adding a sentence about the impact of Cl radicals in the non-polluted troposphere (since that is much of their impact on a global scale). This will set up your readers to better interpret the differences you see between sites since they experience more fresh pollution than studies in other regions.

Response: We agree to address the impact of Cl radicals in the non-polluted troposphere.

Changes in the manuscript:

Line 57-59: The net effect of Cl chemistry is typically the depletion of $O_3$ in the remote atmosphere, such as stratosphere (Molina and Rowland, 1974) and remote oceans (Simpson et al., 2015; Wang et al., 2019b), and an increase in $O_3$ production in the polluted troposphere (Riedel et al., 2014; Xue et al., 2015).

**Line 70-71:** I would also cite Simpson et al., 2015 (ACS: Tropospheric Halogen Chemistry: Sources, Cycling, and Impacts https://pubs.acs.org/doi/10.1021/cr5006638) either here or somewhere else in the introduction.

Response: Simpson et al., 2015 has been added in lines 50-51 and 70-71.

Changes in the manuscript:

Line 50-51: Cl is a potent atmospheric oxidant that reacts analogously to hydroxyl radicals (OH) with hydrocarbons (Simpson et al., 2015).

Line 70-71: The production of Cl is determined by the formation and decomposition of Cl precursors such as $ClNO_2$ (Chang et al., 2011; Simpson et al., 2015).

**Line 98-101:** This statement is not an accurate summation of the reference since they conclude that Cl *is* the dominant radical source in the polluted MBL there (at least in the early morning). I would suggest revising it to: "The role of ClNO2 in the radical budget could be more important than that of OH in winter, because OH production is reduced in winter owing to lower concentrations of O3 and H2O vapor in this season. Haskins et al., 2019 recently confirmed that, even when compared to OH, Cl atoms from ClNO2 photolysis can be the dominant early morning radical source and the dominant integrated daily radical source over the polluted marine boundary layer downwind of the northeast US."

Response: We appreciate the revision made by the reviewer and accept it with minor modification of the wording.

Changes in the manuscript:

Line 105-107: These studies found high $ClNO_2$ mixing ratios of up to 7.7 ppbv (Yun et al., 2018) in winter and a contribution of $ClNO_2$ to Cl liberation of up to 83 % (Priestley et al., 2018) in urban Manchester, and that $ClNO_2$ was a more dominant radical source than OH both in the early morning and the whole day in the polluted marine boundary layer downwind of the northeast US (Haskins et al., 2019).

**Lines 109-119:** I would also mention longer NOx lifetimes allows NOx to spread further distances from its local sources during winter. You may also clarify that the variability in seasonal Cl availability you reference is unique to the NCP.

Response: We have added the information on NOx lifetime in lines 109-119. We have also clarified that the seasonal variability in chloride sources is unique to East Asia in the revised manuscript.

Changes in the manuscript:

Line 111-113: Lower temperatures in winter shift the $N_2O_5$-$NO_3$ equilibrium to the $N_2O_5$ side (Brown et al., 2003) and increase the $\gamma(N_2O_5)$ on aerosols (Bertram and Thornton, 2009). Besides, $NO_x$ has longer lifetimes in winter compared with summer due to less abundant OH radical in winter and its slower reaction rate with OH (Kenagy et al., 2018).

Line 121-124: The availability of aerosol $Cl^-$ also varies in winter and summer. More $Cl^-$ is emitted due to coal burning in winter (McCulloch et al., 1999; Fu et al., 2018). However, in places like East Asia, the winter monsoon brings air masses from the interior of the continent, thereby suppressing the transport of sea salt to inland areas.

**Line 118-119:** This line is quite a strong statement. I suggest reframing it, particularly since the sites examined in this work are subject to quite substantial changes in the precursor conditions. (e.g. "Because of the competing trends and variability in chemical precursors to N2O5 and ClNO2, it is not clear whether ClNO2 formation is always more prevalent during winter compared to summer, particularly in regions that experience large variability in the conditions of the advected air masses they experience.")

Response: We agree with the comment and have reframed lines 118~119. We appreciate the reviewer for suggesting revisions and have made modifications to it.

Changes in the manuscript:

Line 118-119: Because of the contrasts in the availability of aerosol chloride and the variability in meteorology and $NO_x$ emissions that affect the $N_2O_5$ chemistry, it is not clear whether $ClNO_2$ formation is more prevalent in winter.

At some point in the introduction, there needs to be some discussion about our understanding of ClNO2 formation in regions subject to fresh pollution (e.g. lots of fresh NO) verses in aged polluted air masses verses in clean air. All the results in the paper can be explained with what we already know about ClNO2 formation in different types of air masses- but you need to set the reader up for what to expect in each different type of air mass before you get to the results. I think it will set up the discussions of why the trends at the Wangdu site look as they do in a way that will be useful to readers less familiar with the formation process of ClNO2.

Response: We have added more discussion on the $ClNO_2$ formation in different air masses. We suggest adding such discussion at the end of line 107.

Changes in the manuscript:

Line 107: …and a contribution of $ClNO_2$ to Cl liberation of up to 83 % (Priestley et al., 2018). $ClNO_2$ usually exhibits higher concentrations in aged and polluted air masses than in clean air and in regions subject to significant fresh NO emissions (Wang et al., 2016; Wang et al., 2017c; Osthoff et al., 2018).

**Table 1:** I think the "Site Categories" are misleading/unclear. The discussion of each site is excellent & builds a great picture of the conditions experienced, but this table does not summarize those well. On line 158-159 you state that the Wangdu site experiences heavy pollution from coal burning and road traffic. In the results section (lines 285-288) you state that both the Wangdu and Beijing sites experience high NOx and low O3. However, Wangdu is categorized as a "rural" site,

while Beijing is categorized as an "urban" site. While the Wangdu site is certainly more remote than the upwind Beijing site, the category of "rural" is typically used to describe low CO/NOx conditions and urban used to describe places with more CO/ NOx. I suggest either recategorizing Wangdu as "remote polluted" and Beijing as "upwind urban polluted" or adding a column to the table with the average daily NOx and O3 concentrations observed in each observation period. I think the latter might be more useful (because then you could tell that the sites experience rather different pollution conditions in the different seasons). Finally categorizing Mt. Tai as a "mountain" site seems redundant and non-descriptive- perhaps a "remote residual layer" site or "remote clean" site. The abstract and conclusion where these categorical descriptors are used should also be updated (lines 25, 541)

Response: Thanks for the comments and suggestions. Now we recategorize Wangdu as "polluted rural", Beijing as "urban", and Mt. Tai as "polluted lower troposphere". We have also added $NO_x$ and $O_3$ concentrations in Table 1 in the revised manuscript.

Changes in the manuscript:

Line 24-25: This study presents measurements of $ClNO_2$ and related compounds at urban, polluted rural, and polluted lower tropospheric sites in the winter of 2017–2018 over the North China Plain (NCP).

Line 36-37: The daytime-averaged chlorine radical (Cl) production rates ($P$(Cl)) from the daytime $ClNO_2$ were 0.17, 0.11, and 0.12 ppbv h$^{-1}$ at the polluted rural, urban, and polluted lower tropospheric sites, respectively…

Line 541-542: Observations of $ClNO_2$ and related species were conducted at urban, polluted rural, and polluted lower tropospheric sites in the winter of 2017–2018 in the NCP.

Table 1:

| Location/ Coordinate | Site category | Season | Observation period | $NO_x$ (ppbv) | $O_3$ (ppbv) |
|---|---|---|---|---|---|
| Wangdu (38.66 °N, 115.25 °E) | polluted rural | Winter[1] | 9-31 December 2017 | 83.2±81.3 | 4.7±4.5 |
| | | Summer[2] | 21 June to 9 July 2014 | 18.3±11.8 | 37.8±26.2 |
| Beijing (40.04 °N, 116.42 °E) | Urban | Winter[1] | 6 January to 1 February 2018 | 35.6±37.4 | 14.5±11.5 |
| | | Early summer[3] | 24 April to 31 May 2017 | 22.4±18.3 | 27.2±20.6 |
| Mt. Tai (36.25 °N, 117.10 °E) | Polluted lower troposphere | Winter to early spring[1] | 7 March to 8 April 2018 | 2.4±2.0 | 65.1±14.1 |
| | | Summer[4] | 24 July to 27 August 2014 | 3.1±3.0 | 77.8±20.1 |

**Lines 224-244:** A citation for the rate constants used in the box model for each of these equations is needed for reproducibility.

Response: We have added citation for the rate constants $k_1$, $k_i$, $k_{NO+NO3}$, and the equilibrium constant $K_{eq}$.

Changes in the manuscript:

Line 225-226: $P(NO_3)$ was calculated using Eq. (1), where $k_1$ represents the rate constant of Reaction R9 (Atkinson and Lloyd, 1984).

Line 231-233: where $k_i$ is the rate constant for a specific VOC + NO₃ reaction (Atkinson and Arey, 2003) and $k_{NO+NO_3}$ represents the rate constant for Reaction R11 (DeMore et al., 1997). The ambient concentrations of NO₃ were estimated by assuming that NO₃ and N₂O₅ were in dynamic equilibrium (DeMore et al., 1997).

**Line 230-257:** Some discussion about k(NO3) is needed. What VOCs were used? If you did not measure the full suite of VOCs then you would underestimate k(NO3) in Eq. 2 thereby impacting the calculated uptake of N2O5 and yield of ClNO2. A statement about the uncertainty arising from this is needed at minimum.

Response: (1) About k(NO₃). We agree with the suggestion and have added more information about k(NO₃) in line 230-257. The VOCs used to calculate k(NO₃) include non-methane hydrocarbons measured by GC but do not include OVOCs, which was stated in section 2.3, lines 203~207 in the original manuscript, "Online VOCs measurements were performed by gas chromatography-flame-ionization detection/mass spectrometry (GC-FID/MS; Chromatotec Group) at the Beijing site (Zhang et al., 2017) and Wangdu site (Zhang et al., 2020). At Mt. Tai, we used canisters to collect air samples, which were analyzed using GC-FID/MS." As the contribution of OVOCs to k(NO₃) is known to be minor, we do not expect a major underestimation of k(NO₃). We have added a sentence to explain the uncertainty of k(NO₃).

(2) N₂O₅ uptake and ClNO₂ yield. The N₂O₅ uptake coefficient is estimated using the steady state method, in which the VOCs data is not used. So, the calculation of N₂O₅ uptake and ClNO₂ yield is not affected by the accuracy of k(NO₃) calculation.

Changes in the manuscript:
Line 228-229: k(NO₃) during the night was calculated using the measured mixing ratios of NO and VOCs which include non-methane hydrocarbons that can be measured by GC (section 2.3). As most OVOCs react with NO₃ at much slower rates compared to those with hydrocarbons especially alkenes (Atkinson and Arey, 2003), the OVOCs were not included in the calculation of k(NO₃). Nonetheless, the k(NO₃) might be slightly underestimated here.

**Figure 1**: This figure could be improved by making the various y-axis limits consistent across all sites when possible. I see no reason why the jNO2 axis can't be consistent across all panels with a max value at 8*10-3 (as it is on later figures). I would also label the site not with just their name but their "category" e.g. Wangdu: remote polluted). If visibility of the data is impacted, I suggest highlighting the axis differences at minimum.

Response: Thanks for the suggestions. We have now labelled the site with the category information. As Figure 1 presents the key observation results of this study, we hope to clearly exhibit the variability of the species presented here. So, we prefer to keep the various y-axis limits and highlight the axis difference in this figure. We also prefer to use $12*10^{-3}$ s$^{-1}$ as the axis limit of jNO₂ at the Mt. Tai site because the jNO₂ value can occasionally reach above $10*10^{-3}$ s$^{-1}$, e.g., on 18 March 2018.

Changes in the manuscript:

Figure 1:

[Figure]

(a) Wangdu (polluted rural)

(b) Beijing (urban)

(c) Mt. Tai (polluted lower troposphere)

**Figure 2:** Again, this figure could be improved if the y-axis limits were consistent when possible, allowing for easier comparisons. I also think it would be interesting to see the winter and summer data on the same plot rather than split up as it is. Perhaps by using color to denote winter verses summer rather than chemical species/ axis and a translucent shading.

Response: We agree with the suggestion to use consistent y-axis limits for the same species observed in the same season. Besides, we have added an inserted figure to better exhibit the variability of each species when necessary. But we prefer to split up the winter and summer data on different plots, otherwise the figure would look too busy if the winter and summer data are merged. Also, the shaded areas which indicate the 10th and 90th percentiles show important information here, particularly for $ClNO_2$ in Wangdu in winter (Fig. 2a). The 90th percentiles of $ClNO_2$ at 13:00 local time reach 450 pptv due to the presence of significant noontime $ClNO_2$ on several days. Such information would be lost without using shaded areas.

Changes in the manuscript:

Figure 2:

[Figure]

**Lines 310-350:** There is a really important discussion here about the role that shifting NO concentrations plays in driving the difference in the winter and summer measurements at Wangdu and Beijing. However, this important information is not currently communicated in any of the figures of the paper, but easily could be given the observations made.

Response: We have added more content about the relationship of ClNO$_2$ and NO/NO$_x$. Please see below for details.

*(This may be beyond the scope of the authors at this state in revision... But I think it would be interesting to see how ClNO2 formation compared in the winter vs summer observations but grouped by daily peak NO (or NOx) concentrations or daily averaged NO (or NOx) conditions during the day and during the nighttime. (e.g. a bar chart showing ClNO2 concentrations on the y axis, grouped by NO concentrations on the x axis with a bar for winter right next to a bar for summer in each of the NO groupings along the x axis.*

Response: We have drawn a figure in which ClNO$_2$ concentrations are grouped by NO or NOx, respectively, in each field campaign. The mixing ratios of ClNO$_2$, NO and NOx presented here are nighttime values averaged for the whole observation period. As ClNO2 formation mostly occurs during the night, we have only plotted the nighttime relationship of ClNO$_2$ versus NO or NOx, but not include the daytime relationship as suggested by the reviewer.

Changes in the manuscript:

[Figure]

**Figure 3.** The relationship between nighttime levels of ClNO₂ and grouped NO (a, b, and c) and NOₓ (d, e, and f) mixing ratios in the winter (green color) and summer (purple color) campaigns. The difference in the scale of ClNO₂ in Fig. 3c and Fig. 3f is caused by statistic factors, since only 10th to 90th percentile of ClNO₂ data is shown here.

We have also added some sentences to describe this figure in the main text.

Line 321: The mixing ratios of ClNO₂ were mostly low (< 200 pptv) during the night. The relationship between nighttime levels of ClNO₂ and grouped NO and NOₓ concentrations is shown in Fig. 3. ClNO₂ showed higher levels when the NO mixing ratios were below 10 ppbv and NOₓ mixing ratios ranged 10 ~ 20 ppbv (Fig. 3a, d).

Line 342-344: Consequently, N₂O₅ mixing ratios frequently accumulated to elevated levels, exceeding 0.4 ppbv on 10 of the 26 observation nights, and the mixing ratio of ClNO₂ was mostly below 0.4 ppbv. Nighttime levels of ClNO₂ in winter Beijing were higher when NO mixing ratios ranged 0 ~ 10 ppbv and NOₓ mixing ratios ranged 20 ~ 50 ppbv (Fig. 3b, d).

Line 353-354: Elevated mixing ratios of ClNO₂ (i.e., above 0.5 ppbv) were frequently recorded at the Mt. Tai station in winter. Nighttime levels of ClNO₂ were slightly higher when NO levels were below 0.5 ppbv (Fig. 3c) and showed a positive correlation with NOₓ levels (Fig. 3f).

*If the winter/summer average diurnal profiles were combined on Figure 2 in to only 3 panels, you could then show such a figure in panel d, e, & f for each site. It would show how the distribution of NOx as NO vs. NO2 changed between seasons, as well as likely organize why you see more ClNO2 when there is less NO, but more NO2…*

Response: We find it difficult to compress Figure 2 to only 3 panels, as it is necessary to keep the shading area (10th and 90th percentiles) of each species. So, we have decided to insert a new figure about ClNO2 vs NO (NOx) relationship between Figure 2 and Figure 3.

*If that didn't organize well, even showing ClNO2 concentrations grouped by the calculated rate of production of NO3 would communicate the main message of the paper while explaining the average diurnal profiles in a visual way. )*

Response: As we have decided to add the figure about ClNO2 vs. NO (NOx), we prefer not to add another figure to show the ClNO2 concentrations grouped by P(NO3).

**Figure 3:** This would be a great summary figure to have in the literature. However, there appear to be several missing measurements of ClNO2/N2O5, particularly those over the US. The ones off the top of my head which are missing are as follows, but I would encourage the authors to ensure they have included all measurements to date, as I expect there at least a few other observations that are missing…

Faxon et al., 2015: https://www.mdpi.com/2073-4433/6/10/1487

Haskins et al., (2018): https://agupubs.onlinelibrary.wiley.com/doi/full/10.1029/2018JD028786

McDuffie et al., 2019: https://acp.copernicus.org/articles/19/9287/2019/

Jeong et al., (2019) : https://acp.copernicus.org/articles/19/12779/2019/

Response: Thanks for suggesting these additional papers. We have added them. We also searched the literature again and added two more reference: Osthoff et al. (2018). https://acp.copernicus.org/articles/18/6293/2018/, and Sommariva et al. (2018), https://rmets.onlinelibrary.wiley.com/doi/full/10.1002/asl.844.

Changes in the manuscript:
Line 386: 1. (Mielke et al., 2011; Mielke et al., 2016; Osthoff et al., 2018).
Line 386-387: 2. (Osthoff et al., 2008; Faxon et al., 2015)
Line 393: 17. (Brown et al., 2006; Brown et al., 2007; Haskins et al., 2018).
Line 395: 19. (Bannan et al., 2015; Sommariva et al., 2018)
Line 395: 21. (Edwards et al., 2013; Wild et al., 2016; McDuffie et al., 2019).
Line 396: 22. (Jeong et al., 2019).
Figure 2:

[Figure]

**Line 401-408:** These are potentially the most important results of the paper and I'd like to see the few sentences describing them be communicated more clearly. E.g. Explicitly explain why k1 is lower (wintertime temperatures, etc). Explicitly explain why you have lower [NO2] *[O3] in winter (e.g. less photochemical production of O3, more NO in winter titrating available O3, despite longer NO2 lifetimes in winter).

Response: We have added more discussion on $P(NO_3)$ by explicitly explaining $k_1$ and $[NO_2]*[O_3]$. We have also added a new table in SI to show the details of $P(NO_3)$ calculation at each site (see below).

Also while you give avg NOx conditions of the sites during winter there is no discussion about what those are during summer at these sites. Adding that info to Table 1 as previously suggested and perhaps in the paragraphs where the site results are individually introduced would set this discussion up better.

Response: As responded earlier, we have added the $NO_x$ and $O_3$ data in Table 1. When we individually introduce the observation results at each site in section 3.1, we hope to focus on the $ClNO_2$ data in winter seasons. It would be difficult to implement the comparison of $NO_x$ levels between winter and summer in this part. So, we prefer to only add the $NO_x$ and $O_3$ data in Table 1 but do not implement more descriptions to the text.

Changes in the manuscript:

Line 402-403: The lower $P(NO_3)$ in winter was caused by both lower $k_1$ and lower $[NO_2] \times [O_3]$ in winter (see Eq. 1). The lower $k_1$ in winter is caused by lower temperature in winter, while the lower $[NO_2] \times [O_3]$ in winter is mainly caused by less photochemical production of $O_3$ and more NO that consumes the available $O_3$ in winter (Table S5).

Table S5. Comparison of the influencing factors of $P(NO_3)$ in the winter and summer campaigns.

| Place/Season | T (K) | $k_1$ (cm$^3$/molecule/s) | $NO_2$ (ppbv) | $O_3$ (ppbv) | $[NO_2] \times [O_3]$ (molecule$^2$/cm$^6$) | $P(NO_3)$ (ppbv/h) |
|---|---|---|---|---|---|---|
| Wangdu | | | | | | |
| Winter | 271.8±3.2 | (1.5±0.2)×10$^{-17}$ | 34.1±13.0 | 4.7±4.5 | (0.9±0.7)×10$^{23}$ | 0.20±0.15 |
| Summer | 298.1±3.4 | (3.2±0.3)×10$^{-17}$ | 16.8±9.7 | 37.8±26.2 | (2.5±2.1)×10$^{23}$ | 1.34±1.09 |
| Beijing | | | | | | |
| Winter | 270.3±3.7 | (1.4±0.2)×10$^{-17}$ | 27.9±19.1 | 14.5±11.5 | (1.5±1.0)×10$^{23}$ | 0.28±0.18 |
| Early summer | 295.3±4.5 | (3.0±0.4)×10$^{-17}$ | 23.6±13.6 | 27.2±20.6 | (2.9±2.4)×10$^{23}$ | 1.36±1.27 |
| Mt. Tai | | | | | | |
| Winter to early spring | 277.1±5.8 | (1.8±0.3)×10$^{-17}$ | 2.0±1.7 | 65.1±14.1 | (0.8±0.6)×10$^{23}$ | 0.21±0.16 |
| Summer | 289.8±2.1 | (2.6±0.2)×10$^{-17}$ | 3.1±3.2 | 77.8±20.1 | (1.5±1.5)×10$^{23}$ | 0.56±0.55 |

**Figure 4:** Similar to my comments about Figure 2, I wonder if it would be useful to also show on panel A, not just the mean in the winter verses in the summer, but also with the winter and summer measurements at each site separated into two different NOx (or NO) regimes (e.g. fresh pollution present verses not?

Response: We appreciate the reviewer for this suggestion but find it difficult to make such a figure. As the concentration of NO and NO₂ have large variations among the sites and seasons, it is difficult to have a uniform definition of the high-NOx regime and the low-NOx regime at each site.

Also, on panel B would it be possible to show the fraction of loss from NO verses VOCs with hatching? In the text it should also be stated that the loss to VOCs is potentially underestimated (See prior comments).

Response: Agreed. We now show the fraction of $NO_3$ loss from NO. As responded in prior comments, we have clarified the potential underestimation of $k(NO_3)$ in the method part.

Changes in the manuscript:

Figure 4:

[Figure]

**Lines 439-449:** You do not mention if you looked at how IH2O- was changing during these daytime peaks? I suspect its not a problem given the infield calibrations, but it is worth adding a sentence to mention you'd check that as well.

Response: We have checked the $IH_2O^-$ signal and found no abnormal changes during the daytime peaks of $ClNO_2$.

Changes in the manuscript:

Line 446-448: The background signals of $ClNO_2$ were checked when its daytime peaks in concentrations were observed, and no increase in the background was found. We also checked the signal of primary ions ($IH_2O^-$) and found no abnormal changes when $ClNO_2$ concentrations showed daytime peaks.

**Figure 7:** Why is HCHO photolysis not included? I expect it to be a large contributor to the radical budget in at least some of these sites in these periods… Additionally, there's evidence that the presence of Cl radicals oxidizing VOCs enhances the production of HCHO and therefore OH concentrations so the differences between the with and without ClNO2 cases would be underestimated without considering the contributions from HCHO as well? At least some discussion is needed as to why HCHO is not considered part of the radical budget of OH.

Response: In Figure 7a-c, we intend to compare the primary production of OH and Cl radicals. As

HCHO photolysis is a primary source of HO$_2$ but not OH, we did not include HCHO here. In fact, HCHO has already been considered in our box model. In Wangdu and Mt. Tai, HCHO and other OVOCs were measured by offline sampling and post-campaign analysis. As OVOCs were not measured in Beijing in this study, we adopted the concentration of OVOCs measured in previous studies in Beijing. We have added the information on the OVOCs measurements in the method section.

Changes in the manuscript:

Line 206-207: At Mt. Tai, we used canisters to collect air samples, which were analyzed using GC-FID/MS. In Wangdu and Mt. Tai, oxygenated volatile organic compounds (OVOCs) samples were collected on DNPH-coated sorbent cartridges followed by post-campaign analysis using high performance liquid chromatography.

Line 268: …OVOCs and VOCs (Section 2.3) were constrained every hour. As OVOCs were not measured in Beijing in this study, we adopted the concentration of OVOCs measured in previous studies in winter Beijing (Gu et al., 2019; Qian et al., 2019)

The following literature are added to the reference part.

Gu, Y., Li, Q., Wei, D., Gao, L., Tan, L., Su, G., Liu, G., Liu, W., Li, C., and Wang, Q.: Emission characteristics of 99 NMVOCs in different seasonal days and the relationship with air quality parameters in Beijing, China, Ecotoxicology and environmental safety, 169, 797-806, 2019.

Qian, X., Shen, H., and Chen, Z.: Characterizing summer and winter carbonyl compounds in Beijing atmosphere, Atmospheric Environment, 214, 116845, 2019.

**Lines 514 –522:** I'd like to see these results compared to results from other papers that have done this sort of analysis (e.g. Riedel et al., 2012, Young et al., 2014, Haskins et al., 2019, etc.). The conditions of the sites analyzed in this work are sufficiently different from other in the literature (e.g. more fresh NO more coal burning, more aerosol SA, way more HONO) that contrasting your results to those paper's results provides novel insights into the variability of ClNO2 production/ importance to the radical budget during winter in different chemical regimes. Also, Young et al., 2014 (https://acp.copernicus.org/articles/14/3427/2014/ ) is relevant to this work & I'd suggest adding it as a citation.

Response: We agree the suggestion to compare our results to previous studies. We prefer to make such comparison elsewhere but not in line 514-522. Here in line 514-522, we discuss the proportion of VOCs oxidized by Cl and the enhancement of ROx due to Cl, while Riedel et al. (2012), Young et al. (2014), and Haskins et al. (2019) do not present their results in that way. Instead, we suggest comparing the Cl production rate in this study with that of Riedel et al. (2012) and Haskins et al. (2019). We now cite Young et al. (2014) in the introduction part.

Changes in the manuscript:

Line 507-508: The winter $P$(Cl) peak in Wangdu (Fig. 8a, 0.46 ppbv h$^{-1}$) was twice the summer average value (0.24 ppbv h$^{-1}$) (Tham et al., 2016). The P(Cl) during the daytime peaks of ClNO$_2$ in this study is significantly higher than that in Riedel et al. (2012) (maximum ~0.08 ppbv h$^{-1}$) but slightly lower than that in Haskins et al. (2019) (maximum ~1.3 ppbv h$^{-1}$).

Line 51-54: Cl is highly reactive toward alkanes, with the rate constants of its reactions with alkanes being approximately 10–200 times greater than some of the OH + VOCs reactions (Atkinson and Arey, 2003; Young et al., 2014; Burkholder et al., 2015).

**Lines 529-531:** This is the only time that the measurements presented in this paper are put into context into how much they matter in the context of the globe. I'd like to see this sentiment present in the conclusions/ abstract as well to contextualize the results. I suggest adding a statement like

this to the conclusions between lines 546-547.

Response: We agree suggestion to add a statement to highlight the pollution level encountered in this study. We prefer to insert that statement in line 552, as the major results in this study are presented in lines 542-550 and that statement in line 552 summarizes the above results. We also update the abstract to contextualize the results.

Changes in the manuscript:

Line 38-40: Box model calculations showed that the Cl atoms liberated during the daytime peaks of $ClNO_2$ increased the $RO_x$ levels by up to 27–37 % and increased the daily $O_3$ productions by up to 13–18 %. Our results provide new insights into the $ClNO_2$ processes in the lower troposphere impacted by fresh and intense anthropogenic emissions and reveal that $ClNO_2$ can be an important daytime source of Cl radicals under certain conditions in winter.

Line 552: Vertical measurements of the concentrations of $ClNO_2$ and related compounds are needed to better understand the distribution and impact of these species in the lower troposphere. Compared to the previous studies in the clean troposphere or in more aged air masses, our results provide new insights into $ClNO_2$ formation in the region affected by fresh and intense anthropogenic emissions.

**Lines 541:** Again, I don't think its appropriate to categorize the sites as "rural" given their polluted conditions (change to "remote polluted"). Update this line to reflect changes in its "categorization" in Table 1 and in the Abstract.

Response: We recategorize the sites as follows. Wangdu: polluted rural. Beijing: urban. Mt. Tai: polluted lower troposphere. As have responded in the previous comment, we have also updated the category in the text and Table 1.

**Technical Corrections**

**Line 84:** Bertram et al., 2009 should be cited as Bertram & Thornton 2009 as it only has 2 authors.

Response: Here Bertram et al., 2009 refers to "Bertram, T. H., Thornton, J. A., Riedel, T. P., Middlebrook, A. M., Bahreini, R., Bates, T. S., Quinn, P. K., and Coffman, D. J.: Direct observations of $N_2O_5$ reactivity on ambient aerosol particles, Geophysical Research Letters, 36, L19803, 2009.". (Lines 611~613 in the reference section). As this paper has more than two authors, it is cited as "Bertram et al., 2009".

**Line 103:** This reference should be a citation to Haskins et al., 2018 (Wintertime Gas-Particle Partitioning and Speciation of Inorganic Chlorine in the Lower Troposphere Over the Northeast United States and Coastal Ocean https://agupubs.onlinelibrary.wiley.com/doi/full/10.1029/2018JD028786 ) rather than Haskins et al., 2019 (Anthropogenic control over wintertime oxidation of atmospheric pollutants https://agupubs.onlinelibrary.wiley.com/doi/10.1029/2019GL085498 )

Response: Thanks for the pointing out this. We have changed the citation here to Haskins et al., 2018.

Changes in the manuscript:

Line 101-103 …winter observations of $ClNO_2$ have been conducted on various platforms, including on aircrafts over northern Europe (Priestley et al., 2018) and the eastern US (Haskins et al., 2018).

**Line 142:** "mostly during the *heading* period" should likely be "mostly during the *heating* period"

Response: We are sorry for the typo here and have changed "heading" to "heating".

Changes in the manuscript:

Line 141-143: …the observations were made mostly during the heating period during which coal is intensively used.

We went through the manuscript and made additional minor changes shown as follows.

1. Line 187-188: Gas-phase mixtures of $NO_2$ and $O_3$ produced $N_2O_5$ for $N_2O_5$ calibration.

2. Line 226: Some analytical metrics were calculated from the observation data.

3. Line 226: $k(NO_3)$ during the night was calculated using the measured mixing ratios of NO and non-methane hydrocarbons that can be measured by GC (section 2.3).

4. However, the $\gamma(N_2O_5)$ in winter was systematically lower than that in summer (Fig. 6b), which indicated slower $N_2O_5$ loss in winter. A previous field study in winter Beijing also reported small values of $\gamma(N_2O_5)$, ranging < 0.001 to 0.02 (Wang et al., 2020).

Reference: Wang, H., Chen, X., Lu, K., Tan, Z., Ma, X., Wu, Z., Li, X., Liu, Y., Shang, D., and Wu, Y.: Wintertime $N_2O_5$ uptake coefficients over the North China Plain, 65, 765-774, Science Bulletin, 2020.

5. Line 440: Distinct peaks in $ClNO_2$ concentrations were observed on 3–4 days in each campaign, as shown in Fig. 7 displaying one case at each site.

**Author response to review of "Winter observations of ClNO2 in northern China: Spatiotemporal variability and insights into daytime peaks" from Referee #2**

**Men Xia[1], Xiang Peng[1], Weihao Wang[1,8], Chuan Yu[1,2], Zhe Wang[6], Yee Jun Tham[7], Jianmin Chen[4], Hui Chen[4], Yujing Mu[5], Chenglong Zhang[5], Pengfei Liu[5], Likun Xue[2], Xinfeng Wang[2], Jian Gao[3], Hong Li[3], and Tao Wang[1]**

**General Comments:**

This manuscript describes the measurements of ClNO2 and N2O5 at three different locations on the North China Plain (NCP) between 2017 and 2018 and assesses their resulting impact on the radical budget. The three locations are to be representative of an urban, a rural and a mountaintop location. The pollution levels at the rural location are more typical of what one might expect at semi-urban to urban locations so this could be better categorized (discussed further below). The authors show the novel finding of higher ClNO2 in the summer than winter seasons with daytime peaks. The authors demonstrate that the decreased wintertime ozone production coupled with increased loss of NO3 to fresh NO emissions as well as dry wintertime conditions result in lower ClNO2 mixing ratios in the wintertime vs the summertime. An assessment of N2O5 uptake coefficients supports this. This study illustrates that under certain conditions ClNO2 can be an important daytime source of Cl radicals. I feel the authors should be a little more forward in their abstract and conclusion in emphasizing the reasons for the lower wintertime ClNO2 than summer and not just wording it as the "observations". This is an important finding as these measurements were performed in more polluted conditions than many of the ClNO2 measurements in the literature. There are a few places (outlined below) where a little more detail would be helpful to give confidence in measurements without simply citing other publications. Their analysis is well reasoned and consistent with the observations. Overall, the paper is well written and the content is suitable for publication in Atmospheric Chemistry and Physics after addressing the following points.

Response: we appreciate the reviewer for the positive comments and helpful suggestions which have

been addressed in the revision. The reviewer's comments are shown in black font followed by our responses and changes in the manuscript shown in blue and red, respectively. The corrections are also marked as red color in the revised manuscript. Please note that the line numbers mentioned below refer to the original submission (line numbers in the revised version has changed).

We have also revised the abstract and conclusion as suggested by the reviewer.

Changes in the manuscript:

Line 38-40: Box model calculations showed that the Cl atoms liberated during the daytime peaks of $ClNO_2$ increased the $RO_x$ levels by up to 27–37 % and increased the daily $O_3$ productions by up to 13–18 %. Our results provide new insights into the $ClNO_2$ processes in the lower troposphere impacted by fresh and intense anthropogenic emissions and reveal that $ClNO_2$ can be an important daytime source of Cl radicals under certain conditions in winter.

Line 550-552: Vertical measurements of the concentrations of $ClNO_2$ and related compounds are needed to better understand the distribution and impact of these species in the lower troposphere. Compared to the previous studies in the clean troposphere or in more aged air masses, our results provide new insights into $ClNO_2$ formation in the region affected by fresh and intense anthropogenic emissions.

Below is the response to each specific comment.

**Specific Comments:**

**Table 1**: A column showing the ranges of NOx and O3 observed at each of locations would be useful. Showing it summarized here would give the reader a simple indication of the ranges observed at each site. As mentioned before I do not really believe that the categorization of Wangdu site as rural is appropriate given the pollution levels described in the text. Perhaps polluted rural or remote polluted would work.

Response: Thanks for the suggestions. We have added $NO_x$ and $O_3$ concentrations in Table 1 and revised the site categories throughout the manuscript. Now we recategorize Wangdu as "polluted rural", Beijing as "urban", and Mt. Tai as "polluted lower troposphere".

Changes in the manuscript:

Table 1:

| Location/ Coordinate | Site category | Season | Observation period | $NO_x$ (ppbv) | $O_3$ (ppbv) |
|---|---|---|---|---|---|
| Wangdu (38.66 °N, 115.25 °E) | polluted rural | Winter[1] | 9-31 December 2017 | 83.2±81.3 | 4.7±4.5 |
| | | Summer[2] | 21 June to 9 July 2014 | 18.3±11.8 | 37.8±26.2 |
| Beijing (40.04 °N, 116.42 °E) | Urban | Winter[1] | 6 January to 1 February 2018 | 35.6±37.4 | 14.5±11.5 |
| | | Early summer[3] | 24 April to 31 May 2017 | 22.4±18.3 | 27.2±20.6 |
| Mt. Tai (36.25 °N, 117.10 °E) | Polluted lower troposphere | Winter to early spring[1] | 7 March to 8 April 2018 | 2.4±2.0 | 65.1±14.1 |
| | | Summer[4] | 24 July to 27 August 2014 | 3.1±3.0 | 77.8±20.1 |

**P5 L189**: Were these multi-point calibrations or simply span checks? I believe from the SI they were multi-point but it would be helpful to clarify.

Response: Multi-point calibrations of $N_2O_5$ and $ClNO_2$ were performed once in the Mt. Tai campaign. The information has now been added to the main text and SI.

In addition, we would like to clarify that $ClNO_2$ sensitivities were found not affected by RH (see the revised figure below).

Changes in the manuscript:

Line 188-189: The synthetic $N_2O_5$ was converted to $ClNO_2$ by passage through a humidified NaCl slurry for $ClNO_2$ calibration. Multi-concentration calibrations of $N_2O_5$ and $ClNO_2$ were performed once in the Mt. Tai campaign (Fig. S6. The dependence of the $N_2O_5$ sensitivities on ambient RH was tested once in each campaign and used to calibrate the $N_2O_5$ data (Fig. S4). $ClNO_2$ sensitivities were found not affected by RH (Fig. S4b). Single-concentration calibrations of $N_2O_5$ and $ClNO_2$ were performed every $1 - 2$ days, which showed stable sensitivities of $N_2O_5$ and $ClNO_2$ (Text S1 and Fig. S5).

SI, section S1.2: For example, a normalized sensitivity of $1.3\times10^{-5}$ $pptv^{-1}$ of $N_2O_5$ indicates that the sensitivity of $N_2O_5$ is 1.3 Hz/pptv in the presence of $10^5$ Hz of $I(H_2O)^-$ signals. The normalized sensitivities of $ClNO_2$ (($0.9 - 1.8) \times 10^{-5}$ $pptv^{-1}$) and $N_2O_5$ (($1.3 - 2.2) \times 10^{-5}$ $pptv^{-1}$) are stable within each campaign (Fig. S5).

[Figure]

Figure S6. Multi-concentration calibration of $N_2O_5$ and $ClNO_2$ conducted in the Mt. Tai campaign in March 2018.

**P5 L190**: Were these backgrounds only conducted once daily? This seems rather infrequent as many CIMS groups zero their instruments on a significantly more frequent cycle to capture instrument background variability, which can be significant depending on the compound of interest. If they were only done once daily was it always at the same time of day? This should be stated.

Response: The background testing of Q-CIMS was performed once daily at different time in each day. We found that the background signals of $N_2O_5$ and $ClNO_2$ were constant at different time of the day and were much lower than their ambient signals. So, we decided to measure the background signals of $N_2O_5$ and $ClNO_2$ once a day in order to obtain more ambient data.

Changes in the manuscript:

Line 191-192: Background detections of $N_2O_5$ and $ClNO_2$ were conducted by passing ambient air through glass wool once a day at different time. The background signals of $N_2O_5$ (3.3 – 7.7 pptv) and $ClNO_2$ (1.0 – 7.5 pptv) were stable and independent of the time of the day (Fig. S7). The detection limits of $N_2O_5$ and $ClNO_2$ were 6.9 – 7.3 pptv and 3.8 – 5.3 pptv ($3\sigma$ in 5 minutes), respectively (Tabls S2).

[Figure]

Figure S7. Background signals of $N_2O_5$ and $ClNO_2$ in the winter field campaigns over the NCP.

**P5 L192**: How stable was the I(H2O)- signal during the campaign?

Response: The I($H_2O$)$^-$ signal was fairly stable during the campaigns, as can be seen from the average value and standard deviation of I($H_2O$)$^-$ signal shown below.

It's unclear to me whether or not the authors (I couldn't seem to find it in the Xia et al 2019 paper either) added water vapour to the IMR or if the I(H2O)- peak was simply a result of ambient humidity. If it was added, it should be stated and how much.

Response: We did not add water vapor to the IMR of Q-CIMS, so the variation of I($H_2O$)$^-$ signal was a result of change in ambient humidity.

What were the typical count rates for this peak? I ask only because I know some versions of the THS CIMS have a preamp that can saturate around 200-250 kHz and thus some of the changes in ambient humidity may not be captured.

Response: The average count rates for I($H_2O$)$^-$ signal was $(4.2\pm0.9)\times10^4$ in Wangdu, $(3.4\pm0.8)\times10^4$ in Beijing, and $(4.0\pm0.6)\times10^4$ in Mt. Tai during the winter field studies. There is no saturation problem as the count rates of I($H_2O$)$^-$ signal were much lower than 200 kHz.

**P5 L195**: It is a little unclear to me which sensitivities are for which compounds. Did they both vary between 0.9-2.2 x 10^-5? For clarity these should be separated, i.e. N2O5 sensitivities varied between a and b, ClNO2 varied between c and d. Also the units of Hz/Hz/pptv should be expressed as pptv^-1.

Response: Thanks for the suggestion. We have now separately introduce the sensitivity of $N_2O_5$ and $ClNO_2$. We have also changed the units of sensitivity from Hz/Hz/ppt to pptv$^{-1}$ in the revised manuscript.

A better way for comparison with much of the CIMS literature would be to multiply by 1E6 normalized counts per second (ncps) removing the exponential and giving the units of ncps/pptv.

Response: We appreciate this suggestion but think that normalizing $N_2O_5$ and $ClNO_2$ signals by $10^6$ is not applicable in this study. The value $10^6$ refers to the sum of $I^-$ + $I(H_2O)^-$ signals where previous CIMS literature normalizes the $N_2O_5$ and $ClNO_2$ signals, while this study normalizes $N_2O_5$ and $ClNO_2$ signals to $I(H_2O)^-$ signals only. So, multiplying the sensitivity by $10^6$ in this study does not facilitate a comparison with previous studies.

Changes in the manuscript:

SI, section S1.2: For example, a normalized sensitivity of $1.3×10^{-5}$ Hz/Hz/pptv of $N_2O_5$ indicates that the sensitivity of $N_2O_5$ is 1.3 Hz/pptv in the presence of $10^5$ Hz of $I(H_2O)^-$ signals. The normalized sensitivities of $ClNO_2$ ($(0.9 - 1.8) × 10^{-5}$ pptv$^{-1}$) and $N_2O_5$ ($(1.3 - 2.2) × 10^{-5}$ pptv$^{-1}$) are stable within each campaign (Fig. S5).

**P6 L212**: Were the ACSM and MARGA sampling from a common inlet? Was there any size selection (e.g. cyclone) on the front end? A line about this would help strengthen the argument that the missing chloride was simply refractory Chloride containing particles.

Response: The ACSM and MARGA instruments did not share a common inlet. The ACSM and MARGA both measured $PM_{2.5}$ compositions. We have added the cutting size, 2.5 um, in the revise manuscript. We stated in the original manuscript that the missing chloride was refractory chloride-containing particles (see line 214-216: "The concentrations of the $NO_3^-$, $SO_4^{2-}$, and $NH_4^+$ measured simultaneously by the MARGA and ACSM were in good agreement, whereas the concentration of $Cl^-$ measured by the ACSM was slightly lower than that measured by the MARGA, which was possibly due to the significant proportion of refractory chloride, e.g., NaCl, present in the aerosols (Xia et al., 2020)."

Changes in the manuscript:

Line 211-212: An aerosol chemical speciation monitor (ACSM, Aerodyne Research Inc.) was utilized at the Wangdu site to monitor the non-refractory components of these ions in $PM_{2.5}$.

**P6 L230**: (1) What VOC's were used in the calculation of kNO3?

Response: The VOCs used to calculate $k(NO_3)$ include non-methane hydrocarbons measured by GC but do not include OVOCs, which were stated in section 2.3, lines 203~207, "Online VOCs measurements were performed by gas chromatography-flame-ionization detection/mass spectrometry (GC-FID/MS; Chromatotec Group) at the Beijing site (Zhang et al., 2017) and Wangdu site (Zhang et al., 2020). At Mt. Tai, we used canisters to collect air samples, which were analyzed using GC-FID/MS."

(2) Was it simply the compounds listed in the table in the supplementary (S3)?

Response: The VOCs used to calculate k(NO₃) are different from those listed in Table S3, as Table S3 also contains OVOCs.

(3) Either way this table should likely be modified (or a separate table created) to give the actual compound names and formulas as opposed to simply showing their model parameter name.

Response: We revised Table S3 by changing the model parameter name to the actual compound name. Besides, we have now clarified that the OVOCs in Beijing were adopted from previous studies in winter Beijing (Gu et al., 2019; Qian et al., 2019)

Changes in the manuscript:

| No. | Parameter | Wangdu[1] | Beijing[1] | Mt. Tai[1] |
|---|---|---|---|---|
| 1 | $PM_{2.5}$ ($\mu g\ m^{-3}$) | 162.23±90.64 | 116.47±69.33 | 66.65±37.11 |
| 2 | RH (%) | 69.22±7.67 | 33.78±14.67 | 85.52±14.43 |
| 3 | Temp (°C) | -1.95±2.77 | 1.43±4.17 | 9.88±1.53 |
| 4 | NO (ppbv) | 87.84±89.48 | 25.62±27.52 | 0.36±0.35 |
| 5 | $NO_2$ (ppbv) | 39.58±7.68 | 37.24±19.28 | 3.83±2.04 |
| 6 | $O_3$ (ppbv) | 3.74±2.71 | 12.37±10.57 | 59.79±8.6 |
| 7 | CO (ppbv) | 3156.92±1240.79 | 1881.1±898.67 | 623.8±213.84 |
| 8 | $SO_2$ (ppbv) | 11.87±3.28 | 6.22±2.25 | 2.13±1.85 |
| 9 | $N_2O_5$ (ppbv) | 0.02±0.01 | 0.07±0.11 | 0.01±0.02 |
| 10 | $ClNO_2$ (ppbv) | 0.33±0.28 | 0.16±0.17 | 0.26±0.21 |
| 11 | HONO (ppbv) | 4.13±2.41 | 1.02±0.46 | 0.13±0.09 |
| 12 | $jNO_2$ ($\times10^{-3}\ s^{-1}$) | 0.59±0.86 | 0.95±1.43 | 0.88±1.23 |
| 13 | Ethane | 16.13±8 | 0.96±0.42 | 3.93±0.45 |
| 14 | Ethene | 1.93±1.42 | 0.43±0.18 | 1.1±0.53 |
| 15 | Propane | 6.48±2.96 | 6.03±0.9 | 1.94±0.52 |
| 16 | Propene | 5.53±4.27 | 2.02±0.81 | 0.15±0.09 |
| 17 | i-Butane | 1.93±0.51 | 0.79±0.28 | 0.39±0.16 |
| 18 | n-Butane | 3.84±1.14 | 1.18±0.69 | 0.74±0.33 |
| 19 | Ethyne | 6.04±3.53 | 0.38±0.36 | 2.1±0.32 |
| 20 | trans-2-Butene | 0.27±0.26 | - | 0.01±0.01 |
| 21 | 1-Butene | 0.77±0.61 | 0.04±0.01 | 0.03±0.02 |
| 22 | cis-2-Butene | 0.14±0.05 | 0.15±0.04 | 0.01±0.01 |
| 23 | Cyclopentane | 2.67±0.7 | 0.02±0 | 0.04±0.02 |
| 24 | i-Pentane | 0.21±0.17 | - | 0.35±0.18 |
| 25 | n-Pentane | 1.53±0.51 | - | 0.25±0.11 |
| 26 | Freon114 | 0.03±0 | - | 0.02±0 |
| 27 | Chloromethane | 0.45±0.06 | - | 1.1±0.31 |
| 28 | 1,3-Butadiene | 0.27±0.23 | 0.04±0.01 | 0.02±0.01 |
| 29 | Bromomethane | 0.01±0 | - | 0.06±0.06 |
| 30 | Freon11 | 0.4±0.04 | - | 0.26±0.01 |
| 31 | 1-Pentene | 0.18±0.15 | 0.02±0.01 | 0.01±0.01 |
| 32 | trans-2-Pentene | 0.14±0.11 | - | - |
| 33 | Isoprene | 0.13±0.12 | 0.01±0 | 0.01±0.01 |

| | | | | |
|---|---|---|---|---|
| 34 | Freon113 | 0.07±0 | 0.07±0 | 0.08±0 |
| 35 | Dichloromethane | 1.64±0.98 | - | 0.45±0.08 |
| 36 | 2-Methylpentane | 0.44±0.15 | - | 0.06±0.03 |
| 37 | 3-Methylpentane | 0.36±0.09 | - | 0.04±0.02 |
| 38 | 2,2-Dimethylbutane | 0.05±0.01 | - | - |
| 39 | 2,3-Dimethylbutane | 0.07±0.03 | - | - |
| 40 | n-hexane | 0.62±0.23 | 0.22±0.05 | 0.09±0.05 |
| 41 | 2-Propenal | 0.03±0.02 | 0.13±0 | 2.91±1.29 |
| 42 | 2,4-Dimethylpentane | - | - | - |
| 43 | Chloroform | 0.66±0.21 | - | 0.32±0.14 |
| 44 | Methyl chloroform | 0.01±0 | - | 0±0 |
| 45 | Carbontetrachloride | 0.15±0.01 | - | 0.11±0.02 |
| 46 | Cyclohexane | 0.18±0.05 | 0.33±0.16 | - |
| 47 | 3-Methylhexane | - | 0.03±0.01 | - |
| 48 | Methyl tert-butyl ether | 0.16±0.05 | - | - |
| 49 | Benzene | 4.25±2.63 | 1.58±0.37 | 0.71±0.19 |
| 50 | 2,2,4-Trimethylpentan | 0.07±0.03 | - | 0.02±0.01 |
| 51 | n-Heptane | 0.24±0.13 | 0.06±0.03 | 0.04±0.02 |
| 52 | Toluene | 3.28±1.58 | 0.61±0.31 | 0.37±0.24 |
| 53 | n-Octane | 0.17±0.1 | 0.06±0.04 | 0.03±0.01 |
| 54 | Ethylbenzene | 0.47±0.25 | 0.14±0.07 | 0.09±0.03 |
| 55 | n-Nonane | 0.13±0.09 | 0.01±0.01 | 0.04±0.02 |
| 56 | m-Xylene | 1.15±0.65 | 0.29±0.12 | 0.1±0.04 |
| 57 | o-Xylene | 0.41±0.28 | 0.11±0.06 | 0.05±0.02 |
| 58 | Styrene | 0.39±0.31 | 0.03±0.01 | 0.01±0.01 |
| 59 | Isopropylbenzene | 0.04±0.02 | - | 0.01±0 |
| 60 | Propylbenzene | 0.04±0.02 | 0.02±0.01 | 0.01±0.01 |
| 61 | n-Decane | 0.09±0.07 | 0.15±0.07 | 0.02±0.01 |
| 62 | 1,3,5-Trimethylbenzene | 0.06±0.04 | 0.02±0.01 | 0.01±0.01 |
| 63 | 1,2,4-Trimethylbenzene | 0.16±0.12 | 0.06±0.02 | 0.02±0.01 |
| 64 | 1,2,3-Trimethylbenzene | - | 0.1±0.03 | - |
| 65 | Methane | 2000.00[4] | 2000.00[4] | 2044.15±30.31 |
| 66 | Formaldehyde | 3.54±1.03 | 3.18±0[3] | 4.93±1.53 |
| 67 | Acetaldehyde | 2.93±0.69 | 2.5±0[3] | 2.17±0.52 |
| 68 | Propionaldehyde | 0.41±0.07 | 0.29±0[3] | 0.17±0.07 |
| 69 | Acetone | 2.3±0.75 | 2.57±0[3] | 5.11±1.7 |
| 70 | Butyraldehyde | 0.85±0.14 | 0.17±0[3] | 1.5±0.47 |
| 71 | Benzaldehyde | 0.18±0.03 | 0.16±0[3] | 0.11±0.03 |
| 72 | n-Pentanal | 0.27±0.04 | 0.04±0[2] | 0.19±0.04 |
| 73 | Hexanal | 0.12±0 | 0.16±0[2] | 0.13±0.04 |

[1] 24-h average values ± standard deviations are shown here. The units of VOCs and OVOCs are

ppbv. "-" indicates that the parameter is not constrained in the model.
[2] The mixing ratio of this species is adopted from Gu et al. (2019).
[3] The mixing ratio of this species is adopted from Qian et al. (2019).
[4] The mixing ratio of this species is adopted from Tan et al. (2917).

(4) Depending on the completeness of the VOC species measured, kNO3 could very well be underestimated.

Response: As the contribution of OVOCs to $k(NO_3)$ is known to be minor, we do not expect a major underestimation of $k(NO_3)$ (Atkinson and Arey, 2003). We have added a sentence to explain the uncertainty of $k(NO_3)$ due to the incompleteness of the VOC species measured.

Changes in the manuscript:

Line 229-230: $k(NO_3)$ during the night was estimated using the measured mixing ratios of NO and non-methane hydrocarbons that can be measured by GC (section 2.3). As most OVOCs react with NO$_3$ at much slower rates compared to those with hydrocarbons especially alkenes (Atkinson and Arey, 2003), the OVOCs were not included in the calculation of $k(NO_3)$. Nonetheless, the $k(NO_3)$ might be slightly underestimated here.

(5) A table summarizing the rate constants used (it could be in placed in the supplementary) would also be helpful or at a minimum a citation to the rate constants used.

Response: We prefer to add a citation to the rate constants used in the method section.

Line 231-233: where $k_i$ is the rate constant for a specific VOC + NO$_3$ reaction, which is adopted from Atkinson and Arey (2003) and $k_{NO+NO_3}$ represents the rate constant for Reaction R11, which is from (DeMore et al., 1997). The ambient concentrations of NO$_3$ were estimated by assuming that NO$_3$ and N$_2$O$_5$ were in dynamic equilibrium (DeMore et al., 1997).

**P8 L268**: Is the assumption of a constant 2 ppm CH4 mixing ratio reasonable for both the high and low coal burning seasons?

Response: As coal burning is a significant source of CH$_4$, the concentration of CH$_4$ should be different in high and low coal burning seasons. We tried to search the literature but could not find a comparison of CH$_4$ levels in high and low coal burning seasons in northern China. We have acknowledged in the revised manuscript that the adoption of the summer CH$_4$ concentration (Tan et al., 2017) for our winter study may underestimate the CH$_4$ level.

    In addition, we would like to clarify that we performed the box model simulation only for the winter campaigns (high coal burning seasons) in northern China but not summer campaigns (low coal burning seasons).

I don't have a feeling for what the difference would be and I'm little surprised it wasn't measured as part of the list of VOC's.

Response: It is a pity that CH$_4$ was not measured in Wangdu and Beijing but only measured in Mt. Tai (2044±30 ppbv in average). As Wangdu and Beijing are closer to fresh emission sources compared with Mt. Tai, the CH$_4$ concentration (assumed to be 2000 ppbv) might be underestimated in Wangdu and Beijing. We have now acknowledged the uncertainty of CH$_4$ in the revised manuscript. We think the uncertainty of CH$_4$ should not significantly affect the budget of OH, HO$_2$, and RO$_x$.

Changes in the manuscript

We assumed the mixing ratio of CH$_4$ to be constant at 2000 ppbv in Wangdu and Beijing (Tan et al., 2017). As Wangdu and Beijing are closer to fresh emission sources compared with Mt. Tai, the CH$_4$

concentrations may be underestimated in Wangdu and Beijing and cause slight uncertainties to the RO$_x$ budgets.

**P7 L484**: I feel like a plot showing the NOx (or even just NO) data would be of value. Perhaps Figure 1 could be modified to add this as a trace. While it is well described in the text it would be of value to the reader to see the trends overlaid with the other time traces.

Response: We agree that adding NOx data to Figure 1 would make it more informative to the readers. However, we have a concern about the size of the Figure 1. Adding the NOx data at each site would result in three additional panels, which makes it too busy. So, we can only show the most indispensable data here, which we think is N$_2$O$_5$, ClNO$_2$, O$_3$, and jNO$_2$. As an alternative, NOx and additional data have been shown in Figure S6. Following the reviewer's suggestion, we now label the category of each site on Figure 1 (which is also suggested by RC1).

**Figure 2**: It would be easier to visualize the winter/summer comparison contrast with the plots overlaid on each other. If the axis could also be consistent across the measurement locations, it would make it easier for the reader to discern the differences between the measurement locations.

Response: We have revised Fig. 2 to use consistent y-axis limits for the same species observed in the same season. We have add an inserted figure to better display the variability of each species when necessary. The revised Figure 2 has been shown in our response to RC1. We prefer to split up the winter and summer data on different plot. The figure would look too busy when the winter and summer data are merged. Also, the shaded areas which indicate the 10$^{th}$ and 90$^{th}$ percentiles show important information, particularly for ClNO$_2$ in Wangdu in winter (Fig. 2a). The 90$^{th}$ percentiles of ClNO$_2$ at 13:00 local time reach 450 pptv due to the presence of noontime ClNO$_2$ on several days. Such information would be lost without using the shaded areas.

**P9 L321**: The presence of elevated ClNO2 with high NO levels suppressing N2O5 formation is a really important observation from this work and should probably be highlighted more than it is. The authors should consider including a figure so that the reader can better visualize this. Perhaps one with a couple of panels showing two or 3 different elevated ClNO2/NO events.

Response: We already showed representative cases of elevated daytime ClNO$_2$ events in Figure 6 and more cases in Figure S9~S11 in the original manuscript. As shown in our responses to RC1, we have now added a new figure between Figure 2 and Figure 3 to show the nighttime relationship between ClNO$_2$ and grouped NO and NOx.

**Figure 3**: I really like this figure but there are perhaps a couple of references missing. I know of at least 1

(McDuffie, E. E., Womack, C. C., Fibiger, D. L., Dube, W. P., Franchin, A., Middlebrook, A. M., Goldberger, L., Lee, B. H., Thornton, J. A., Moravek, A., Murphy, J. G., Baasandorj, M., and Brown, S. S.: On the contribution of nocturnal heterogeneous reactive nitrogen chemistry to particulate matter formation during wintertime pollution events in Northern Utah, Atmos. Chem. Phys., 19, 9287–9308, https://doi.org/10.5194/acp-19-9287-2019, 2019.)

I encourage the authors to go back through the literature to make sure that no other measurements have been missed.

Response: Thanks for pointing out a missing reference. We have added it together with others suggested by another reviewer and a few more based on our literature search.

Changes in the manuscript:

Line 386: 1. (Mielke et al., 2011; Mielke et al., 2016; Osthoff et al., 2018).
Line 386-387: 2. (Osthoff et al., 2008; Faxon et al., 2015)
Line 393: 17. (Brown et al., 2006; Brown et al., 2007; Haskins et al., 2018).
Line 395: 19. (Bannan et al., 2015; Sommariva et al., 2018)
Line 395: 21. (Edwards et al., 2013; Wild et al., 2016; McDuffie et al., 2019).
Line 396: 22. (Jeong et al., 2019).
The revised Figure 3 has been shown in our response to RC1.

**P15 L527**: Any idea where the source of BrCl might be? Was Br2 observed during any of the campaigns?

Response: BrCl observed here might be originate from coal burning, reactive uptake of HOBr on chloride-containing particles, and activation of bromide by nitrate photolysis. The mean mixing ratio of $Br_2$ was 4 ppt in Wangdu. More details can be found in our recent paper (Peng et al., 2020). As reactive bromine species is not the focus of this study, we prefer not to elaborate it here.

**Supplementary S1.1**

It is not true that there is no known interference for N2O5 at m/z 235. Veres et al (2020) have shown that in the marine boundary layer that hydroperoxymethyl thioformate (a DMS oxidation product) does overlap with N2O5 in the I- CIMS spectrum at m/z 235. While this interference is not likely to be present in this case, some discussion of it is warranted.

Response: Thanks for pointing out the potential interference of $N_2O_5$ signal. We agree to add more discussion and prefer to put this part in SI.

Changes in the manuscript:

SI, section S1.1: As for $N_2O_5$, some field measurements with higher mass resolutions showed no interference, e.g., Breton et al. (2018). However, a recent study revealed that hydroperoxymethyl thioformate, an oxidation product of dimethyl sulfide (DMS) by OH, does overlap with the $N_2O_5$ signal at 235 a.m.u in their iodide HR-ToF-CIMS (Veres et al., 2020). This interference was negligible at our three sites due to very low daytime signals of 235 a.m.u., typically below 15 pptv by assuming all 235 a.m.u. signals were $N_2O_5$. This result is consistent with anticipated low concentrations of DMS at our inland sites.

Was DMS one of the VOC's measured?

Response: It is a pity that DMS was not measured in these campaigns. We have searched the literature but does not find DMS measurements in polluted inland sites of northern China.

It would also be useful to include a table showing the masses measured with their corresponding integration times to demonstrate the instruments duty cycle.

Response: We have added the instrument duty cycle and integration times of the measured species by adding the table below in SI.

Changes in the manuscript:

Line 186: An example of the mass spectrum is shown in Fig. S2. The integration time of the signals recorded by the Q-CIMS is shown in Table S1.

Table S1: Integration time of the signals recorded by the Q-CIMS in the winter campaigns.

| Mass to charge ratio | Integration time (ms) | Mass to charge ratio | Integration time (ms) |
|---|---|---|---|
| 62 | 287 | 210 | 287 |
| 145 | 587 | 217 | 287 |

| | | | |
|---|---|---|---|
| 163 | 290 | 222 | 290 |
| 165 | 287 | 223 | 287 |
| 173 | 287 | 234 | 286 |
| 174 | 287 | 235 | 1237 |
| 178 | 287 | 241 | 287 |
| 179 | 287 | 243 | 287 |
| 192 | 287 | 245 | 287 |
| 197 | 287 | 254 | 287 |
| 199 | 288 | 280 | 287 |
| 207 | 287 | 287 | 287 |
| 208 | 1239 | 289 | 288 |
| 209 | 287 | 291 | 287 |
| Total integration time | | 10.2 s | |

Citation: Veres, P. R., Neuman, J. A., Bertram, T. H., Assaf, E., Wolfe, G. M., Williamson, C. J., Weinzierl, B., Tilmes, S., Thompson, C. R., Thames, A. B., Schroder, J. C., Saiz-Lopez, A., Rollins, A. W., Roberts, J. M., Price, D., Peischl, J., Nault, B. A., Møller, K. H., Miller, D. O., Meinardi, S., Li, Q., Lamarque, J.-F., Kupc, A., Kjaergaard, H. G., Kinnison, D., Jimenez, J. L., Jernigan, C. M., Hornbrook, R. S., Hills, A., Dollner, M., Day, D. A., Cuevas, C. A., Campuzano-Jost, P., Burkholder, J., Bui, T. P., Brune, W. H., Brown, S. S., Brock, C. A., Bourgeois, I., Blake, D. R., Apel, E. C., and Ryerson, T. B.: Global airborne sampling reveals a previously unobserved dimethyl sulfide oxidation mechanism in the marine atmosphere, Proceedings of the National Academy of Sciences, 117, 4505-4510, 10.1073/pnas.1919344117, 2020.

Response: We appreciate the reviewer to recommend this reference to us. It has been added in the part of discussing the potential interference of $N_2O_5$.

**Figure S6:** I find the number of colours used on this plots a little overwhelming.

Response: We appreciate this comment but prefer to use the current configuration of color on Figure S6. Overall, it is easier to differentiate the measured species or parameters by using more colors. See below for detailed explanations.

Perhaps you could recycle the same two colours per stacked plot as there is only one trace per axis?

Response: We recycled the same two colors (red and blue) in the original version for the T – RH panel and $N_2O_5$ – $ClNO_2$ panel. The colors of $O_3$ – $jNO_2$ panel are set to keep consistent with Figure 1 (purple and orange, respectively). Other panels show more than two traces, i.e., the NO, $NO_x$, $NO_y$ panel and the $Cl^-$, $NO_3^-$, $SO_4^{2-}$, $NH_4^+$, and $PM_{2.5}$ panel. So, more colors are needed to differentiate these species.

**Figure S7**: I think these panels would be more informative/useful if the plots were binned by RH as opposed to simply being coloured by RH. It would more strongly demonstrate the higher corrrelations at high RH values. If this resulted in too many plots the results could be summarized in a table with a single exemplar plot.

Response: Thanks for your suggestion. We prefer to keep the current format of Figure S7, as it conveys the same message whether the figure is binned by RH or colored by RH. The impact of RH on the ratio of $ClNO_2/N_2O_5$ is clearly demonstrated by the current plotting.

**Technical Corrections:**

**P4 L142**: I assume this should read "during the heating period"?

Response: We are sorry for the typo here and have changed "heading" to "heating".

Changes in the manuscript:
Line 141-143: …the observations were made mostly during the heating period during which coal is intensively used.

**P10 L363-364**: The wording appears reversed; I'm assuming it's a simple translation issue. The decrease in SO2 should be due to the reduced effect of coal-fired power.

Response: We are sorry for the inappropriate expression here and have revised the wording.

Changes in the manuscript:

Line 363-365: The reduced coal-fired power generation caused the continued decrease in $SO_2$ emissions during 2014-2018 and less transport of emissions from the ground to the Mt. Tai site.

We went through the manuscript and made additional minor changes shown as follows.

1. Line 187-188: Gas-phase mixtures of $NO_2$ and $O_3$ produced $N_2O_5$ for $N_2O_5$ calibration.

2. Line 226: Some analytical metrics were calculated from the observation data.

3. Line 226: $k(NO_3)$ during the night was calculated using the measured mixing ratios of NO and non-methane hydrocarbons that can be measured by GC (section 2.3).

4. However, the $\gamma(N_2O_5)$ in winter was systematically lower than that in summer (Fig. 6b), which indicated slower $N_2O_5$ loss in winter. A previous field study in winter Beijing also reported small values of $\gamma(N_2O_5)$, ranging < 0.001 to 0.02 (Wang et al., 2020).
Reference: Wang, H., Chen, X., Lu, K., Tan, Z., Ma, X., Wu, Z., Li, X., Liu, Y., Shang, D., and Wu, Y.: Wintertime $N_2O_5$ uptake coefficients over the North China Plain, 65, 765-774, Science Bulletin, 2020.

5. Line 440: Distinct peaks in $ClNO_2$ concentrations were observed on 3–4 days in each campaign, as shown in Fig. 7 displaying one case at each site.